# Cryo-EM structures define ubiquinone-10 binding to mitochondrial complex I and conformational transitions accompanying Q-site occupancy

Injae Chung [1], John J. Wright [1,4], Hannah R. Bridges [1,4], Bozhidar S. Ivanov[1,4], Olivier Biner[1,3], Caroline S. Pereira [2], Guilherme M. Arantes [2] & Judy Hirst [1✉]

Mitochondrial complex I is a central metabolic enzyme that uses the reducing potential of NADH to reduce ubiquinone-10 ($Q_{10}$) and drive four protons across the inner mitochondrial membrane, powering oxidative phosphorylation. Although many complex I structures are now available, the mechanisms of $Q_{10}$ reduction and energy transduction remain controversial. Here, we reconstitute mammalian complex I into phospholipid nanodiscs with exogenous $Q_{10}$. Using cryo-EM, we reveal a $Q_{10}$ molecule occupying the full length of the Q-binding site in the 'active' (ready-to-go) resting state together with a matching substrate-free structure, and apply molecular dynamics simulations to propose how the charge states of key residues influence the $Q_{10}$ binding pose. By comparing ligand-bound and ligand-free forms of the 'deactive' resting state (that require reactivating to catalyse), we begin to define how substrate binding restructures the deactive Q-binding site, providing insights into its physiological and mechanistic relevance.

[1] MRC Mitochondrial Biology Unit, University of Cambridge, The Keith Peters Building, Cambridge Biomedical Campus, Hills Road, Cambridge CB2 0XY, UK. [2] Department of Biochemistry, Instituto de Química, Universidade de São Paulo, Av. Prof. Lineu Prestes 748, São Paulo, SP 05508-900, Brazil. [3] Present address: Institute of Plant and Microbial Biology, University of Zurich, Zollikerstrasse 107, 8008 Zürich, Switzerland. [4] These authors contributed equally: John J. Wright, Hannah R. Bridges, Bozhidar S. Ivanov ✉email: jh@mrc-mbu.cam.ac.uk

Mitochondrial complex I (NADH:ubiquinone oxidor-eductase) is an intricate ~1 MDa multimeric membrane-bound complex that is essential for mito-chondrial metabolism[1,2]. It comprises 14 catalytic core subunits that are conserved across all kingdoms of life, and up to 31 supernumerary subunits that contribute to its stability, reg-ulation, and/or biogenesis[3,4]. As an entry point into the electron transport chain (ETC), complex I is a key contributor to oxidative phosphorylation, mitochondrial homeostasis, and redox balance. Specifically, it couples electron transfer from NADH to ubiqui-none (Q) to proton translocation across the inner mitochondrial membrane, building the proton motive force ($\Delta p$) to drive ATP synthesis. Due to its central roles in metabolism, complex I is implicated in ischaemia-reperfusion (IR) injury[5] and its dys-functions lead to neuromuscular and metabolic diseases[6].

Two biochemically characterised, physiologically relevant states, the 'active' and 'deactive' resting states described initially by Vinogradov and coworkers[7,8], have previously been identified by electron cryomicroscopy (cryo-EM) on mammalian complex I[3,9,10]. They are distinguished by subtle domain movements and con-formational changes at the Q-binding site and in adjacent membrane-domain subunits[3,9–13]. In the absence of substrates and at physiological temperatures, mammalian complex I relaxes gra-dually from the ready-to-catalyse active resting state into the pro-found deactive resting state, with a partially unstructured Q-binding site[3,7–10,14,15]. The deactive state forms spontaneously during ischaemia (when lack of $O_2$ prevents ETC turnover). Then, when NADH and ubiquinone are added/replenished (for example, upon reperfusion), it slowly reactivates and returns to catalysis. During reperfusion the deactive state protects against IR injury by mini-mising complex I-mediated production of reactive oxygen species by reverse electron transfer[13,16,17]. Cryo-EM studies of mammalian complex I have also identified a third state, for which the density map lacks information in key regions of the enzyme, which we suggested previously to arise from complex I in the first stages of dissociation[3,11].

The Q-binding site in complex I is an unusually long and heterogeneous channel. Advances in cryo-EM have allowed identification of several inhibitors[12,18–22], adventitious detergents[23], and substrate analogues[19,20,24,25] bound in the site, whilst native Q species ($Q_9$ and $Q_{10}$, with nine and ten isoprenoid units, respectively) have predominantly been observed with their Q-headgroups close to the channel entrance[25–28]. There is cur-rently only one report of complex I containing a fully-bound $Q_{10}$ molecule, which is for porcine complex I in the mammalian respirasome[25]. As expected, the structure shows the fully-bound $Q_{10}$ spanning the entirety of the channel, but, consistent with observations of the short-chain substrate analogues decylubiqui-none (dQ)[19,20,24,25] and piericidin A[12,19], it does not exhibit the expected dual-ligation of the two redox-active carbonyls on the Q-headgroup. Biochemical data[29] and molecular dynamics simulations[30–32] both indicate that, for redox-coupled proton transfer, the headgroup must be ligated between two proton-donor ligating partners (His59$^{NDUFS2}$ and Tyr108$^{NDUFS2}$) at the tip of the channel.

Here, we describe the structures of several states of bovine complex I reconstituted into phospholipid nanodiscs with exo-genous $Q_{10}$. The nanodiscs provide a native membrane-like environment and eliminate potential artefacts from the detergent micelle typically present in cryo-EM analyses. Using cryo-EM, we resolve five distinct structures at global resolutions up to 2.3 Å, including one with a $Q_{10}$ molecule occupying the full length of the Q-binding channel. By comparing the structures of substrate/ligand-bound and apo (substrate/ligand-free) forms of both the active and deactive states, as well as a structure of the poorly characterised third state, we probe substrate/ligand-driven conformational changes in the Q-binding site and the physiolo-gical and catalytic relevance of each state.

## Results

**Reconstitution of bovine complex I into nanodiscs.** Complex I was purified from bovine heart mitochondria in the detergent *n*-dodecyl β-D-maltoside (DDM)[10,33]. Then, to exchange the detergent micelles for nanodiscs, it was reconstituted in a mixture of synthetic phospholipids, $Q_{10}$, and the membrane scaffold protein MSP2N2[34], using a protocol derived from that for pre-paring complex I proteoliposomes[21,33,35]. The nanodisc-bound complex I (CxI-ND) was then isolated by size-exclusion chro-matography (from proteoliposomes, protein-free nanodiscs, free MSP2N2, and residual detergent) and shown to be monodisperse (Supplementary Fig. 1a). Following addition of CHAPS (3-((3-cholamidopropyl) dimethylammonio)-1-propanesulfonate) and asolectin to dissociate the scaffold proteins and provide a larger hydrophobic phase, the sample used for cryo-EM exhibited 86.0 ± 0.1% (17.5 ± 0.4 μmol min$^{-1}$ mg$^{-1}$) of the piericidin-sensitive NADH:dQ oxidoreductase activity of the DDM-solubilised enzyme before reconstitution (20.4 ± 0.6 μmol min$^{-1}$ mg$^{-1}$), indicating that the complex I in CxI-ND is highly cata-lytically competent. The intact CxI-NDs (without CHAPS/aso-lectin) displayed very little piericidin-sensitive NADH:dQ activity (2.01 ± 0.10 μmol min$^{-1}$ mg$^{-1}$) indicating that dQ does not exchange effectively in and out of the nanodisc and/or Q-binding site (see also Supplementary Fig. 1b).

**Resolution of three major classes of CxI-ND particles.** Single-particle cryo-EM analyses on a Titan Krios microscope with a K3 detector yielded a total of 343,213 particles (Table 1), which were separated into three major classes by 3D classification in RELION[36], following subtraction[37] of the nanodisc density (Supplementary Fig. 2). By map-to-map comparisons with bio-chemically characterised mouse and bovine structures[3,9,10,12,13] (Supplementary Table 1a), two classes were assigned to complex I in the 'active' resting state (61,658 particles, 2.65 Å resolution) and in the 'deactive' resting state (259,540 particles, 2.28 Å). The third class (22,019 particles, 3.02 Å) matched best to a state proposed earlier to correspond to particles in the first stages of dissociation; here we name it 'state 3' to reassess it without the bias from early suggestions based on 5.60 Å low-resolution data[3,11]. Focused 3D classifications subsequently resolved two sub-states for each of the active and deactive states, providing a total of five distinct maps and models (Table 1 and Supplemen-tary Figs. 2, 3 and 4).

All the established hallmarks for the active state[3,10,12] are present in the active maps/models (Supplementary Fig. 5), including well-defined densities for the loops in NDUFS2, ND3 and ND1 (residues 52–62, 24–55, and 194–217, respectively) that constitute the Q-binding site and for the region of NDUFA9 closest to the membrane, an extended NDUFA5/NDUFA10 interface, and a fully α-helical ND6-transmembrane helix (TMH) 3. [Note we use the human subunit nomenclature throughout for simplicity]. The same is true for the deactive maps/models (Supplementary Fig. 5), where the hallmarks include disordered/ alternate conformations of the above loops, disorder in a short stretch of a loop in NDUFA9 (residues 324–331) adjacent to the disordered loop in ND3, a restricted NDUFA5/NDUFA10 interface, and a π-bulge in ND6-TMH3[3,9,10,13]. Furthermore, NDUFS7 residues 47–51 form a loop in the active state and a β-strand in the deactive state, and the adjacent loop (residues 74–83) is 'flipped over' between the two states, reorientating the hydroxylated conserved Arg77$^{NDUFS7}$. As reported previously[3], the state 3 map (see Supplementary Fig. 6) lacks clear densities for

**Table 1 Cryo-EM data collection, refinement, and validation statistics.**

| Data collection and processing | Data collection I | Data collection II |
|---|---|---|
| Magnification | Grid I / 81,000 | Grid II / 81,000 |
| Voltage (kV) | 300 | 300 |
| Electron exposure (e⁻/Å²) | 40.5 | 40.5 |
| Defocus range (μm) | −1.0 to −2.4 | −1.0 to −2.4 |
| Super-resolution pixel size (Å) | 0.535 | 0.535 |
| Final pixel size (Å) | 0.750 | 0.750 |
| Symmetry imposed | C1 | C1 |
| Initial particle images (no.) | 701,236 | 377,697 |
| Final particle images (no.) | 212,841 | 130,372 |
| Total final particle images (no.) | 343,213 | |

| Classes | Active-$Q_{10}$ EMD-14132 PDB-7QSK | Active-apo EMD-14133 PDB-7QSL | Deactive-ligand[a] EMD-14134 PDB-7QSM | Deactive-apo EMD-14139 PDB-7QSN | State 3 (Slack) EMD-14140 PDB-7QSO |
|---|---|---|---|---|---|
| Final particle images (no.) | 23,449 | 38,205 | 235,957 | 23,583 | 22,019 |
| Map resolution (Å) [FSC threshold: 0.143] | 2.84 | 2.76 | 2.30 | 2.81 | 3.02 |
| Map resolution range (Å) | 2.61-7.02 | 2.49-6.70 | 2.05-3.99 | 2.48-7.16 | 2.61-6.49 |
| Map sharpening B-factor (Å²) | −34 | −42 | −(49-53) | −44 | −54 |
| **Model statistics** | | | | | |
| Initial model (PDB ID) | 7QSD | 7QSD | 7QSD | 7QSD | 7QSD |
| Model resolution (Å) [FSC threshold: 0.5] | 2.85 | 2.73 | 2.23 | 2.73 | 2.91 |
| **Model composition** | | | | | |
| Non-hydrogen atoms | 69,987 | 69,743 | 71,642 | 69,322 | 66,114 |
| Protein residues | 8,287 | 8,283 | 8,259 | 8,198 | 8,015 |
| Ligands | 45 | 42 | 48 | 45 | 32 |
| Waters | 1,060 | 1,024 | 2,774 | 1,096 | 0 |
| **Average B-factors (Å²)** | | | | | |
| Protein | 49.62 | 43.88 | 40.03 | 42.27 | 51.36 |
| Ligand | 50.65 | 44.98 | 42.75 | 43.24 | 50.30 |
| Water | 44.47 | 37.52 | 37.10 | 36.74 | N/A |
| **Root Mean Square deviations** | | | | | |
| Bond lengths (Å) | 0.007 | 0.009 | 0.006 | 0.006 | 0.006 |
| Bond angles (°) | 0.641 | 0.732 | 0.647 | 0.648 | 0.618 |
| MolProbity score | 1.30 | 1.36 | 1.13 | 1.37 | 1.44 |
| All-atom clash score | 3.18 | 3.76 | 2.69 | 3.68 | 4.03 |
| EMRinger score | 4.82 | 5.15 | 6.42 | 4.68 | 4.07 |
| Rotamer outliers (%) | 0.15 | 0.04 | 0.58 | 0.01 | 0.00 |
| **Ramachandran plot** | | | | | |
| Favoured (%) | 96.87 | 96.83 | 97.64 | 96.69 | 96.21 |
| Allowed (%) | 3.07 | 3.13 | 2.34 | 3.28 | 3.75 |
| Outliers (%) | 0.06 | 0.04 | 0.01 | 0.02 | 0.04 |

[a]Data refers to the composite map, except for the map resolution (FSC = 0.143) which comes from the consensus map data. For the consensus map data, see Supplementary Fig. 3.

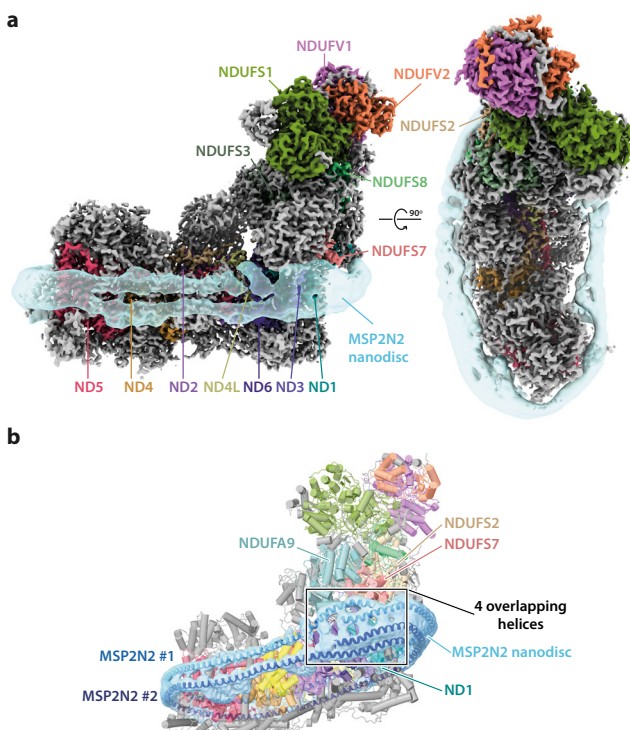

**Fig. 1 Overview of the structure of mitochondrial complex I from *Bos taurus* reconstituted into nanodiscs. a** Side and top views of the cryo-EM densities for the 14 core (coloured) and 31 supernumerary (grey) subunits of $Q_{10}$-bound active complex I at a map threshold of 6.5 are shown with the subtract-refined MSP2N2 nanodisc (transparent sky blue) overlaid in UCSF ChimeraX[51]. **b** Heel view of a representative CxI-ND model (deactive-ligand; cartoon) encapsulated in the subtract-refined MSP2N2 nanodisc map with polyalanine models for the MSP2N2 molecules shown as helices.

the C-terminal half of the ND5 transverse helix and its TMH16 anchor; subunit NDUFA11 (barring a short fragment facing the intermembrane space); and the ~40-residue N-terminus of NDUFS2. These densities remain unclear at the current — much higher — resolution, confirming these elements are disordered or in multiple unresolved conformations. Here, we focus on the occupancy of the Q-binding channel in each state and evaluate the biochemical relevance of the state 3 structure in Supplementary Note 2.

**Characterisation of nanodisc-bound complex I.** In all three states, two belt-like densities representing two MSP2N2 monomers are visible, enveloping the membrane domain in place of the usual detergent:phospholipid micelle (Fig. 1). Two 391 residue-long polyalanine models fit into them well, overlapping as four parallel helices below NDUFA9 (Fig. 1b), a subunit just above the expected membrane surface that has been observed to bind phospholipids[10] and reported as an important feature of the active to deactive transition[38]. The overlapping contrasts with the 'dangling' termini observed for shorter MSPs[39]. Quantitative biochemical analyses showed the CxI-NDs contain an average of 295 phospholipids, sufficient for a layer only one or two deep around the enzyme, and 2.66 $Q_{10}$. Although this equates to ~12 mM $Q_{10}$ in the phospholipid phase, substantially above the $K_M$ value in proteoliposomes[33], there is no 'bulk' bilayer to support quinone diffusion, consistent with the low $dQ/Q_{10}$ reductase activity (Supplementary Fig. 1b). Furthermore, the MSP2N2s wrap tightly around the enzyme like stretched rubber bands, creating direct enzyme-MSP2N2 contacts in some regions

and phospholipid-filled crevices and cavities where there are protein voids in others (Fig. 1 and Supplementary Fig. 7a).

Comparison of the CxI-ND active-state structure with a DDM-solubilised active-state bovine structure [Protein Data Bank (PDB) ID: 7QSD; Electron Microscopy Data Bank (EMDB) ID: 14127][40] revealed no material differences, with convincing map-to-map correlation (0.97) and RMSD values (0.26–0.31 Å for the membrane-bound core subunits, 0.32–0.35 Å overall). For the deactive-state structures, we were able to model substantially more of subunit NDUFA9 than has previously been possible in detergent-solubilised deactive/open structures[3,9,10,13,20,25]. Only residues 324–331, adjacent to the disordered loop in ND3, were not modelled, suggesting that the nanodisc environment is able to contain the disorder that further propagates to residues ca. 186–195, 253–278 and 323–334[13,20,25] in the detergent-solubilised enzyme. It is thus possible that structural changes to NDUFA9 in the membrane-bound deactive state are less extensive than previously supposed. As expected, none of the three DDMs modelled in the active-state bovine reference structure[40] were observed in CxI-NDs, while the total number of phospholipids observed has increased to 47, including all 22 that were modelled in the reference structure[40] (Supplementary Fig. 7b). There is no evidence that 'lateral pressure' tightens the subunit interfaces in the CxI-ND membrane domain, and further comparisons with detergent-solubilised mammalian complex I structures[9,10,12,13,20,21,40] revealed only three very minor structural differences (see Supplementary Note 2).

**$Q_{10}$ bound in the active state.** Following focused 3D classification to resolve heterogeneous density features in the Q-binding site (see Methods and Supplementary Fig. 3), the active class was split into a ligand-bound sub-class with a continuous density matching a $Q_{10}$ molecule spanning the Q-binding channel (henceforth active-$Q_{10}$; 23,449 particles, 2.84 Å resolution) and a substrate-free sub-class with a presumably water-filled cavity (henceforth active-apo; 38,209 particles, 2.76 Å) (Table 1 and Supplementary Fig. 4). The Q-binding site is fully structured in both cases.

In active-$Q_{10}$, the $Q_{10}$ density occupies the entirety of the channel, starting between subunits NDUFS2 and NDUFS7 (with the Q-headgroup adjacent to Tyr108[NDUFS2] and His59[NDUFS2]) then extending, as the isoprenoid chain, along the NDUFS2-NDUFS7 then ND1-NDUFS7 interfaces to exit from ND1. The His sidechain forms a hydrogen bond (H-bond) with the 3-methoxy of the Q-headgroup (3.2 Å), rather than with either of the reactive carbonyls, which are in geometrically unfavourable positions (Fig. 2a). The Tyr sidechain is too distant (>4.3 Å) from the headgroup for a H-bond, but interacts via two mediating water molecules and a water is also H-bonded between the 4-carbonyl and Asp160[NDUFS2]. In an alternative lower-occupancy orientation of the $Q_{10}$, represented by weaker density, the headgroup is flipped by 180° and the first isoprenoid is in a different position (Fig. 2a inset). While the His now interacts with the 2-methoxy the H-bonding pattern is similar. In both cases, attempts to reposition the headgroup to create reactive H-bonds to the Tyr and His without moving the isoprenoid chain out of its density were unsuccessful. A number of waters were also observed adjacent to the isoprenoid chain, stabilised by H-bonding to nearby sidechains, clustered particularly in the more charged protein section around the middle of the isoprenoid chain[33] (Fig. 2b).

To probe further why the $Q_{10}$ appears to have 'paused' in this pre-reactive conformation, we applied molecular dynamics simulations with enhanced sampling to explore the charge-state and flexibility of reactive groups in the Q-binding site (see Methods for

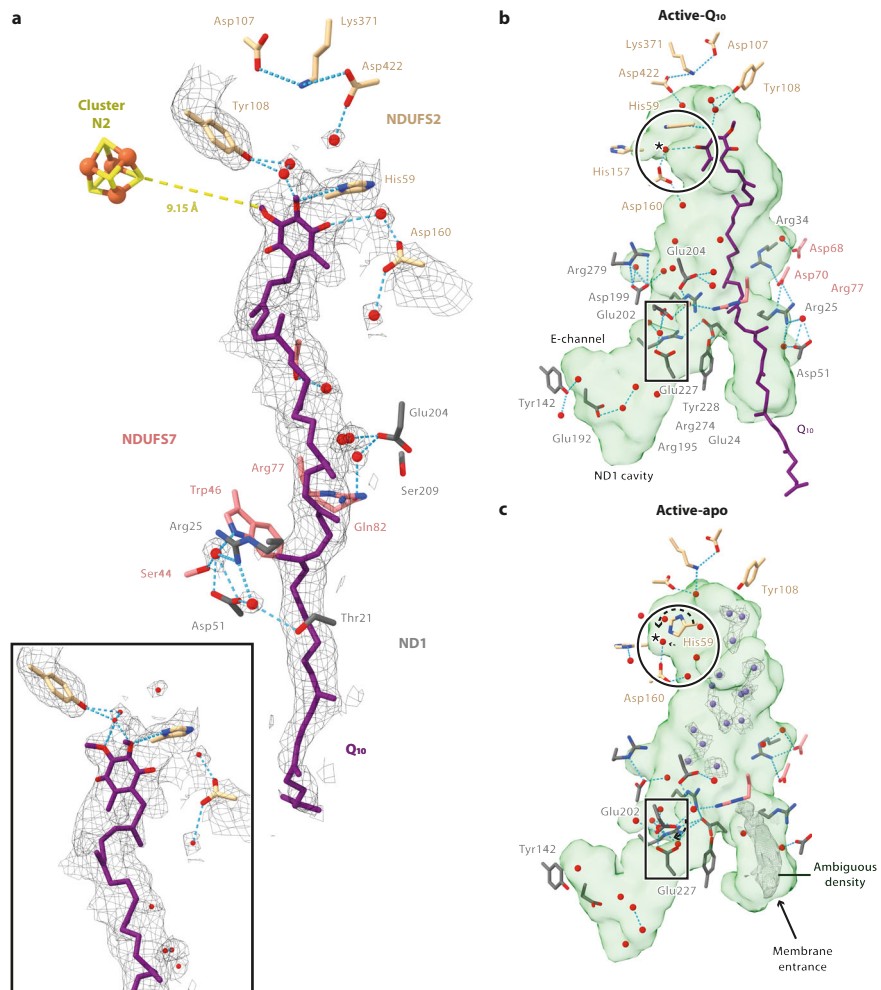

**Fig. 2 Active states of complex I with and without bound Q$_{10}$. a** Cryo-EM densities of Q$_{10}$ and neighbouring water molecules in the active-Q$_{10}$ map at a map threshold of 4.4 are shown together with the polar residues that make H-bonding interactions, as identified using the *hbonds* command in UCSF ChimeraX[51]. The inset shows an alternate conformation of Q$_{10}$ that can be modelled into the density. The independent model-map CC$_{mask}$ values for the primary and secondary Q$_{10}$ poses are 0.72 and 0.70, respectively. Protonatable residues, water molecules (red spheres) and ligands in close proximity to the Q-binding channel and ND1 cavity (the green surfaces identified by CASTp[45]) are shown and labelled in the (**b**) active-Q$_{10}$ and (**c**) active-apo states. Star (*) indicates a water molecule that moves as a result of the rotation of His59$^{NDUFS2}$ (dotted arrow; circled). The unidentified density and densities for the dynamic water molecules (violet spheres) in the active-apo map are shown at a map threshold of 4.0 in UCSF ChimeraX[51].

details). A collective variable (CV) was used to describe the Q-headgroup position along the binding channel, with lower values denoting the headgroup closer to Tyr108$^{NDUFS2}$ (Fig. 3a–d). Three charge-states were tested, all with cluster N2 oxidised but different protonation states for His59$^{NDUFS2}$ and Asp160$^{NDUFS2}$ (Figs. 2a and 3a). The free energy profile obtained for the [AspH + His] charge-state exhibits a well-defined single minimum, indicating stable binding at CV = 23.8 Å (Fig. 3a) with the Q-headgroup position matching that observed in the active-Q$_{10}$ cryo-EM model at CV = 23.6 Å (Fig. 3a). The other two charge-states ([Asp$^-$ + His] and [AspH + HisH$^+$]) show multiple minima, with the lowest at CV = ~25 Å having the Q-headgroup distant from the reactive site. Additional structural properties (Supplementary Fig. 8) also showed significantly better agreement with the active-Q$_{10}$ cryo-EM model for the [AspH + His] charge-state. The [Asp$^-$ + HisH$^+$] charge-state was excluded because the cryo-EM model did not suggest an ionic-pair interaction. Thus, our simulations strongly suggest the experimental structure is in the [AspH + His] charge-state and that the Q-headgroup binding pose at the top of the Q-channel is modulated by the charge-states of nearby groups.

Our active-Q$_{10}$ model was built with the His59$^{NDUFS2}$ dihedral angle $\chi_2 = -95.5°$ (Fig. 2a), but the conformer with a 'flipped' sidechain also fits well. Interestingly, simulations for the [AspH + His] charge-state (Fig. 3e) revealed a flexible His59$^{NDUFS2}$ sidechain with a barrier for the ring to flip of <15 kJ mol$^{-1}$ at the stable binding position (CV = 23.8 Å), consistent with C$_\beta$-C$_\gamma$ bond rotamers with $\chi_2 = -100°$ or +80° being equally populated at the physiological temperature simulated. When the Q-headgroup approaches Tyr108$^{NDUFS2}$ (CV = 22.5 Å; Fig. 3b), the flexibility of His59$^{NDUFS2}$ $\chi_2$ is slightly decreased (Fig. 3e) and a H-bond, which is absent at the stable binding position (Fig. 3c), forms between His59$^{NDUFS2}$-N$_{\epsilon2}$ and the protonated Asp160$^{NDUFS2}$ sidechain (Fig. 3f). It may also re-form when the Q-headgroup dissociates (Fig. 3d), in line with previous simulations on complex I from *T. thermophilus*[31,41]. The H-bond between His59$^{NDUFS2}$-N$_{\delta1}$ and the Q-headgroup 3-methoxy is easily broken (Fig. 3g). In fact, when the Q-headgroup occupies its stable binding position it has sufficient conformational freedom to twist and flip during the physiological-temperature simulations, also visiting configurations compatible with the flipped subpopulation observed in the cryo-EM data (Fig. 2a inset).

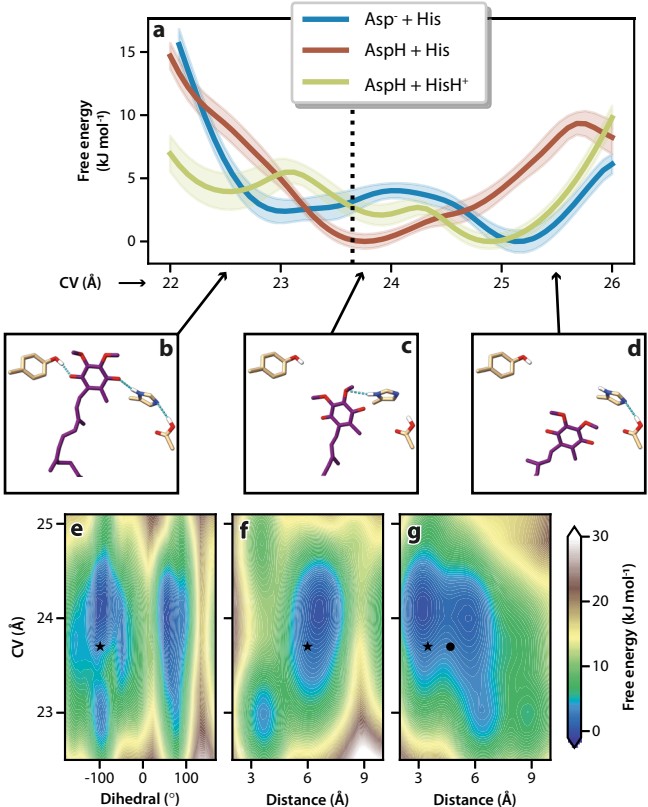

**Fig. 3 Free energy profiles from molecular dynamics simulations.**
**a** Profiles were obtained for three combinations of sidechain protonation with Asp160[NDUFS2] ionised (Asp⁻) or protonated (AspH) and His59[NDUFS2] neutral (His, $N_{\delta 1}$-protonated $\pi$ tautomer) or di-protonated (HisH⁺, both $N_{\delta 1}$- and $N_{\epsilon 2}$-protonated). The statistical uncertainties in the free energies were estimated as 95% confidence intervals by bootstrap analysis and are shown in coloured shadows. The collective variable (CV) describes the Q-headgroup position along the binding channel, as illustrated with His59[NDUFS2], Tyr108[NDUFS2] and Asp160[NDUFS2] sidechains in **b** for CV = 22.5 Å, **c** for CV = 23.8 Å, and **d** for CV = 25.5 Å. The dashed line in **a** shows CV = 23.6 Å measured from the active-Q₁₀ cryo-EM model. Two-dimensional profiles for the [AspH + His] charge-state along the CV coordinate and coloured by free energy are shown in **e** for the His59[NDUFS2] $\chi_2$ dihedral ($C_\beta$-$C_\gamma$ bond torsion), **f** for the distance between the His59[NDUFS2]-$N_{\epsilon 2}$ and Asp160[NDUFS2]-$C_\gamma$ atoms, and **g** for the distance between the Q-headgroup 3-methoxy-O and His59[NDUFS2]-$N_{\delta 1}$ atoms. Symbols correspond to the structural properties observed in cryo-EM models active-Q₁₀ with the primary (star) and flipped (bullet) Q-headgroups.

In the absence of contiguous density attributable to bound Q₁₀ (active-apo structure), we suggest that the discrete densities scattered throughout the Q-binding channel, including in isoprenoid-binding regions, are dynamic water molecules in H-bonding networks (Fig. 2c). Intriguingly, there is also a clear but unidentified density observed at the channel entrance (it may be a Q₁₀ or phospholipid inserted tail-first, but is also consistent with a molecule of MOPS buffer) that appears to separate the water-filled cavity from the hydrophobic membrane (Fig. 2c). In the reactive site, the His59[NDUFS2] sidechain is rotated by ~90° relative to in the active-Q₁₀ structure (Fig. 2c) and stabilised by H-bonding to the carbonyl backbone of Ile423[NDUFS2]; the water molecule between the Q-headgroup and Asp160[NDUFS2] shifts by ~1 Å in response (Fig. 2c). The two waters between Tyr108[NDUFS2] and the Q-headgroup are not resolved in the active-apo sub-state, but we suggest a poorly-resolved density extending

from Tyr108[NDUFS2] represents a network of dynamic water molecules, in place of the headgroup. Apart from His59[NDUFS2], the only difference in residue conformation between the Q-binding sites in the active-Q₁₀ and active-apo structures is at Glu202[ND1]. Although, as may be expected for carboxylates[42], the sidechain densities are not well-resolved, the models suggest that in active-Q₁₀ Glu202[ND1] forms a water-mediated H-bond to Glu227[ND1] whereas in active-apo, Glu202[ND1] has undergone a rotameric shift that reconfigures the H-bonding interactions between the two glutamates. These observations hint that Glu202[ND1] responds to the channel occupancy, and may function as a 'control point' for proton transfer.

**Ligand binding to the deactive state.** Q-binding site loops in subunits NDUFS2, ND1 and NDUFS7 are characteristically disordered in the deactive state[9,10,13] and so further classification of the deactive particles was focused on the whole Q-binding region (Supplementary Fig. 3). Two sub-classes were resolved: an occupied, ligand-bound sub-class (deactive-ligand; 235,957 particles, 2.30 Å resolution) and an apparently unoccupied 'apo' sub-class (deactive-apo; 23,583 particles, 2.81 Å) (Table 1 and Supplementary Fig. 4). In the deactive-apo structure, the Q-binding site loops in NDUFS2, ND3 and ND1 are largely disordered, whereas in the deactive-ligand structure the NDUFS2 and ND1 loops are in ordered conformations different from those in the active state (Fig. 4 and Supplementary Fig. 5). The overall structure remains deactive (with a restricted NDUFA5/NDUFA10 interface, disordered ND3 loop, π-bulge in ND6-TMH3, and deactive NDUFS7 conformation). The restructured NDUFS2-β1-β2 loop (which carries His59) has moved into the space occupied by the Q-headgroup in the active-Q₁₀ structure (Fig. 4c, f), and together with the restructured ND1-TMH5-6 loop, it constricts the Q-binding channel (Fig. 4c). In contrast, in the deactive-apo structure, the disordered loops fail to enclose the channel, which appears as a gaping crevice open to the matrix (Fig. 4d).

The ligand density observed in the deactive-ligand structure displays features consistent with both Q₁₀ and DDM, suggesting a mixed population that could not be separated by focused classification. The DDM used for complex I preparation may have been retained in the channel when the external DDM molecules were removed during reconstitution. The shape of the headgroup density suggests Q₁₀ at low map thresholds but matches two six-membered maltoside rings at higher thresholds, while a long, zigzagged protrusion resembles an isoprenoid chain (Supplementary Fig. 9). Fitting a Q₁₀ molecule into the density reveals just one polar interaction, a H-bond from Arg274[ND1] to the Q-headgroup, while the fitted DDM molecule additionally interacts with His55[NDUFS2] and Glu202[ND1], and an intervening water molecule bridges it to Glu24[ND1] (Fig. 5a). DDM may thus stabilise the deactive state, but whether it also promotes deactivation[23,43] remains unclear. His55[NDUFS2] and Glu202[ND1] are on the displaced NDUFS2-β1-β2 and ND1-TMH5-6 loops, respectively, consistent with their restructuring in the deactive-ligand structure and with continued disorder in the ND3-TMH1-2 loop, which is stabilised in the active state[23] by interaction between His55[NDUFS2] and Cys39[ND3]. Notably, the DDM molecule modelled here differs from the one observed in *Y. lipolytica* complex I[23], which is slightly further into the Q-binding channel with its maltoside rings in different positions (Supplementary Fig. 10g–i). Similarly, the modelled positions of Q₁₀ in plant complex I[27] and porcine complex I[25], Q₉ in *Y. lipolytica* complex I[26], and dQ in closed and open ovine complex I[20] (Supplementary Fig. 10m–p) overlap with the Q₁₀/DDM modelled here, but do not align well. Clearly, the lower section

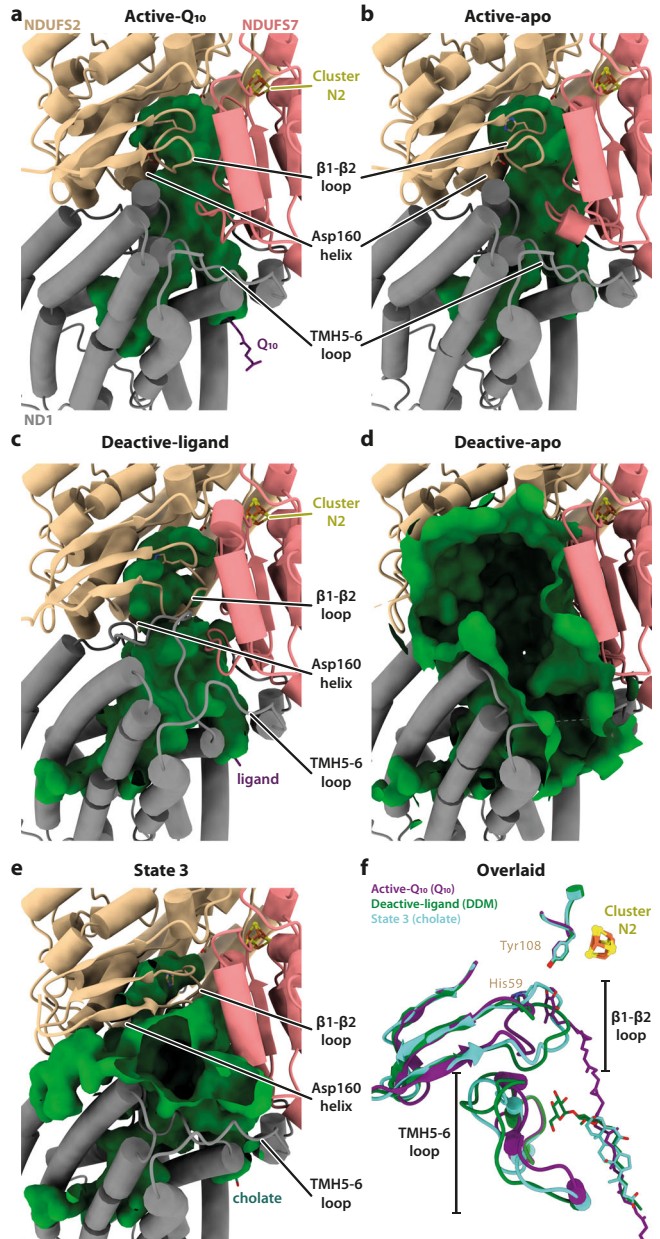

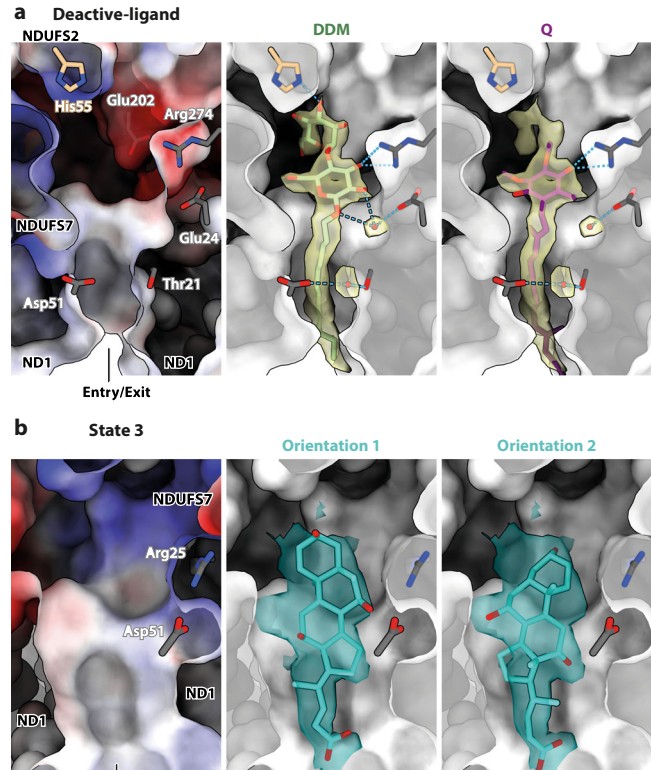

**Fig. 4 Conformations of the Q-binding site loops and the cavities they encapsulate in the five observed states of complex I. a–e** Cartoon representations of subunits ND1, NDUFS2, and NDUFS7 showing varying extents of disorder indicated as cavities (green surfaces) detected by CASTp[45]. Where visible, His59NDUFS2, Tyr108NDUFS2 and Asp160NDUFS2 are shown as sticks. **f** Superimposed structures of the active-$Q_{10}$ (purple), deactive-ligand (green) and state 3 (turquoise) states, showing the NDUFS2-β1-β2 and ND1-TMH5-6 loops in cartoon, and His59NDUFS2 and Tyr108NDUFS2 in sticks.

of the Q-binding channel can accommodate a variety of extended hydrophobic and amphipathic molecules, including substrates, inhibitors and detergents, and their binding may modify surrounding protein structures.

**Ligand binding to state 3.** Following focused classification procedures, all the state 3 particles were retained in a single homogenous class that contains densities for two bound ligands. First, a clear density at the entrance to the Q-binding channel (Fig. 4e, f)

**Fig. 5 Ligands bound at the entrance of the Q-binding site of complex I.** A clipped view of the surface representation of the entrance to the Q-binding channel in the (**a**) deactive-ligand and (**b**) state 3 models [coloured by Coulombic electrostatic potential (left) or by depth, in black and white (middle and right)]. **a** Cryo-EM density of the ligand in the deactive-ligand composite map (see Methods) at a map threshold of 4 (transparent yellow) in UCSF ChimeraX[51], modelled as either a DDM (middle) or a $Q_{10}$ (right). Residues H-bonded to the ligand or surrounding water molecules are shown and labelled. The model-map $CC_{mask}$ values for the DDM and $Q_{10}$ (clipped to $Q_3$) are 0.74 and 0.70, respectively. **b** Cryo-EM density of the ligand in the state 3 map at a map threshold of 4.5 (transparent teal), modelled as a cholate molecule in two different orientations. Residues within H-bonding distance to the ligand are shown and labelled. The model-map $CC_{mask}$ values for the primary and secondary cholate orientations are 0.65 and 0.67, respectively.

is consistent with a cholate molecule (added for the reconstitution) in two orientations (Fig. 5b), in close proximity to either Arg25ND1, Asp51ND1, and Trp46NDUFS7, or Arg274ND1, Thr21ND1, and Tyr228ND1 for polar interactions. As in the deactive-ligand structure, the NDUFS2-β1-β2 and ND1-TMH5-6 (but not ND3-TMH1-2) loops are ordered, but in different conformations to the deactive-ligand or active states, so the shape of the Q-binding channel also differs (Fig. 4e, f). The NDUFS2-β1-β2 loop is translated up the channel relative to both its active and deactive-ligand conformations, bringing the backbone carbonyls of Ala58NDUFS2 and His59NDUFS2 to within H-bonding distance (2.6–2.8 Å) of the Tyr108NDUFS2 hydroxyl and restricting the cavity. The ND1-TMH5-6 loop does not run across the Q-binding channel as in the deactive-ligand state, but is retracted downwards, beyond its conformation in the active state, so the Q-binding site again appears open to the matrix. It is not possible to tell if these changes result from cholate binding or are intrinsic to state 3. Regardless of the physiological and mechanistic relevance of state 3, our structure affirms the flexible nature of the Q-binding site and its ability to accommodate ligands. Second, a

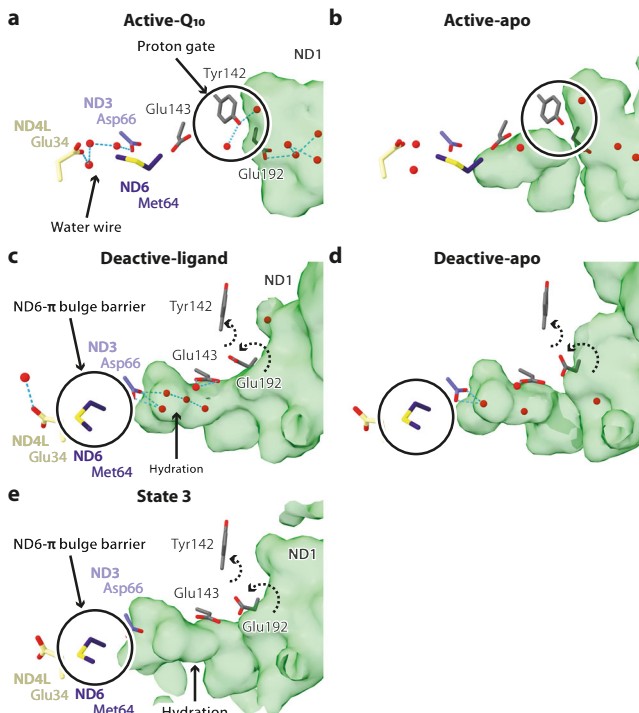

**Fig. 6 The ND1 cavity and the E-channel that connect the Q-site to the first antiporter-like subunit ND2.** The proton pathways from subunits ND1 to ND4L in the (**a**) active-$Q_{10}$, (**b**) active-apo, (**c**) deactive-ligand, (**d**) deactive-apo, and (**e**) state 3 states. Cavities (green surfaces) were identified by CASTp[45] using a 1.4 Å probe. E-channel residues are shown in sticks and labelled. Dashed arrows indicate movement of residues with respect to their conformations in the active states.

clear density for an ordered $Q_{10}$ molecule is observed at the extended interface between subunits ND2 and ND4 (see Supplementary Note 1 and Supplementary Fig. 6a, b) resulting from their opposing rotation and retraction[3,44], and is accommodated by a π-bulge in ND4-TMH6 not present in the active or deactive states. However, the $Q_{10}$ headgroup is ~100 Å away from cluster N2 and the Q-binding channel, which is clearly too far for catalytic electron transfer. Similarly, a non-catalytic $Q_{10}$ has also been proposed to bind in subunit NDUFA9 (outside of the canonical Q-binding site), in the active state of porcine complex I[25]. In our active states, this unorthodox $Q_{10}$-binding site is well ordered and occupied by a phospholipid molecule.

**State-dependent structural features in the ND1 cavity and E-channel.** To probe whether the different Q-site structures propagate changes to the membrane domain, we investigated the structures of the E-channel, which leads from the Q-binding site and the solvent-accessible cavity in ND1 to the first antiporter-like subunit, ND2.

The shapes of the ND1 cavities, surrounded by subunits ND1, ND3, and ND6, were visualised using the CASTp software with a 1.4 Å diameter probe[45] (Figs. 2b, c, 4a–e and 6). In both active-$Q_{10}$ and active-apo, the cavity extends past Glu227$^{ND1}$ and Glu192$^{ND1}$ (Fig. 2b, c) but is blocked from the next E-channel residue (Glu143$^{ND1}$) by Tyr142$^{ND1}$, which may thus act as a 'proton gate' (Fig. 6a, b). Two water molecules, on each side of Tyr142$^{ND1}$, connect the hydrated ND1 cavity to Glu143$^{ND1}$ and Asp66$^{ND3}$. In both deactive-ligand and deactive-apo, the ND1 cavity extends past Glu227$^{ND1}$, Glu192$^{ND1}$, Glu143$^{ND1}$, and up to Asp66$^{ND3}$. The cavity is extended by the straightening of ND1-TMH4, which forces Tyr142$^{ND1}$ to rotate out of the way, shifting

the Glu192$^{ND1}$ sidechain towards Glu143$^{ND1}$ into an electrostatic interaction as described previously[20,23,25,32], and bringing additional hydration (Fig. 6c, d). In this state, Glu143$^{ND1}$ and Asp66$^{ND3}$ are stabilised by two intermediary water molecules H-bonded to the backbone carbonyl of Gly62$^{ND6}$. As in other structures in the deactive[13,23,24] or open[20] state, as well as in simulations[32], the ND6-TMH3 π-bulge wedges Met64$^{ND6}$ into the ~12.5 Å-long Asp66$^{ND3}$-to-Glu34$^{ND4L}$ H-bond network that is continuous in the active state, disrupting it and releasing Glu34$^{ND4L}$ to flip away towards ND2 (Fig. 6c, d). In this region, state 3 matches the deactive state (Fig. 6e). Beyond this region, we did not observe any substantial changes in the membrane domain between the active-$Q_{10}$/-apo and deactive-ligand/-apo states.

## Discussion

The active-$Q_{10}$ structure described here shows an exogenous native ubiquinone substrate inserted fully into the complex I Q-binding channel. However, the Q-headgroup is not bound in the expected reactive state with its 1,4-carbonyls H-bonding to Tyr108$^{NDUFS2}$ and His59$^{NDUFS2}$, ready to receive their protons following electron transfer from cluster N2. Simulations of the reactive state in *T. thermophilus* complex I[30,32] proposed that the 4-carbonyl always H-bonds to Tyr108$^{NDUFS2}$, but the 1-carbonyl may either H-bond or π-stack with His59$^{NDUFS2}$. While H-bonding between the 4-carbonyl and Tyr108$^{NDUFS2}$, but not between the 1-carbonyl and His59$^{NDUFS2}$, was described in a recently published active $Q_{10}$-bound porcine structure[25], our active-$Q_{10}$ structure shows neither of these interactions. Furthermore, previous structures with the $Q_{10}$ analogue dQ and the inhibitor piercidin A1 bound are also not modelled in the expected reactive state (Supplementary Fig. 10a–f). In the ovine 'closed' state supplemented with dQ and NADH[20] the dQ-headgroup is rotated ~35° (in-plane) relative to our primary binding pose so that His59$^{NDUFS2}$ H-bonds to the 4-carbonyl instead of the 3-methoxy, but the 1-carbonyl is ~4.5 Å from Tyr108$^{NDUFS2}$. The dQ-headgroup is 5-6 Å away from Tyr108$^{NDUFS2}$ (Tyr144$^{NDUFS2}$) in *Y. lipolytica* complex I also[24]. In contrast, the dQ-headgroup in a crystal structure of *T. thermophilus* complex I[19] is flipped and rotated out-of-plane, with Tyr108$^{NDUFS2}$ (Tyr87$^{Nqo4}$) H-bonding with both the 4-carbonyl and 3-methoxy but His59$^{NDUFS2}$ (His38$^{Nqo4}$) not interacting. A similarly flipped binding pose was captured in another set of simulations on *T. thermophilus* complex I[41], where the $Q_{10}$-headgroup was stabilised by a H-bond to (protonated) His59$^{NDUFS2}$ but did not engage Tyr108$^{NDUFS2}$. The binding poses observed in two piericidin-bound structures[12,19] match the pose for dQ in the *T. thermophilus* structure[19], although neither recapitulates the currently unique NDUFS2-β1-β2 loop conformation that increases the separation of Tyr108$^{NDUFS2}$ and His59$^{NDUFS2}$ in this dQ-bound bacterial structure.

Here, we showed excellent agreement between our experimentally determined $Q_{10}$ binding pose and predictions from simulations with cluster N2 oxidised and a defined [AspH + His] charge-state for Asp160$^{NDUFS2}$ and His59$^{NDUFS2}$. Our results imply that the protonation and charge states of key active-site residues are intimately connected to the binding pose. Furthermore, in both our structure and simulations, cluster N2 is oxidised. Reduction of cluster N2 is likely a key driver for the changes required to bring the Q-headgroup from the pose we observe into the reactive state, including 'flipping' the Q-headgroup to orientate the 4-carbonyl toward Tyr108$^{NDUFS2}$, shifting ionisation equilibria to protonate His59$^{NDUFS2}$, and changing the conformation of the NDUFS2-β1-β2 loop to bring it into a coordinating position. We suggest that our observed state is structurally equivalent to a 'pre-reactive' state that occurs

naturally on the catalytic cycle, similar to that proposed previously by Teixeira and Arantes[31], in which the $Q_{10}$ pauses before dehydration and ligation of the 1,4-carbonyls finally brings it into the reactive conformation. The active $Q_{10}$-bound porcine structure described recently[25] differs from our structure because there the $Q_{10}$ headgroup is flipped and the 4-carbonyl is H-bonded to Tyr108$^{NDUFS2}$. This structure was further suggested to be in a different charge-state, with His59$^{NDUFS2}$ protonated and a much shorter distance to Asp160$^{NDUFS2}$ than in our structure (1.7 Å vs. 5.0 Å) suggesting an ion-pair interaction that places it in the [Asp$^-$ + HisH$^+$] charge-state. Nevertheless, His59$^{NDUFS2}$ remains unable to coordinate to the 1-carbonyl. Complex I is expected to encounter several different charge-states, including for Asp160$^{NDUFS2}$ and His59$^{NDUFS2}$, during its turnover cycle and therefore the two different $Q_{10}$ binding poses observed may illustrate the interplay of charge-state and Q-headgroup ligation that the system explores prior to entering the reactive conformation.

On the surface of it, our results may alternatively be taken to suggest the possibility of ubiquinone reduction in improperly ligated states, consistent with the considerable (inhibitor-sensitive) NADH:$Q_1$/$Q_2$ activities of the Tyr108$^{NDUFS2}$-equivalent Y144F/W/H mutants in *Y. lipolytica*[29]. However, ubiquinone reduction potentials depend on the ligation environment and binding mode[46], and engaging water molecules around the headgroup as non-specific proton donors appears unlikely in the context of an efficient energy-conserving mechanism. Bound $Q_1$/$Q_2$ have much greater conformational freedom without the long isoprenoid tail to constrain them[25,33,47], so they may exhibit alternative hydration patterns and conformational distribution[31], and become artefactually activated for reduction.

Here, reconstituting complex I into phospholipid nanodiscs with $Q_{10}$ allowed us to resolve substrate/ligand-bound and substrate/ligand-free forms of both the active and deactive states of mammalian complex I, plus a ligand-bound form of state 3. Whether deactive-like states (in which the NDUFS2-β1-β2, ND3-TMH1-2, ND1-TMH5-6 loops and a loop in subunit NDUFS7 are altered/disordered[10,23]) are intermediates in catalysis[20] or pronounced off-cycle resting states[7,8,14,15,25] is currently a major controversy. First, we note the high similarity of our bovine/murine active and deactive states with the closed and open conformations, respectively, of ovine complex I, supported by high map-map correlations, small overall RMSD values (Supplementary Table 1b) and matching structural hallmarks[9,10,20]. Whereas we attribute the deactive state to an off-cycle pronounced resting state, Kampjut and Sazanov[20] proposed that 'open' deactive-like states must form during catalysis for $Q_{10}$ to enter the Q-binding channel, which would then 'close' as it moves to the reactive site. Their model excludes unoccupied active-like/ closed states as catalytic intermediates. However, the simplest explanation for our observation of both the active-$Q_{10}$ and active-apo states, with near-identically structured Q-binding sites (as also observed in several inhibitor-bound structures[12,20–22]), is that Q-binding may occur within active-like/closed conformations. In this case, individual structural elements involved in the deactive transition may move during catalysis, but would not be coordinated to produce the extensive domain-level conformational changes required to generate a full deactive-like state. Simulations of Q-binding/dissociation differ in the extent to which they predict conformational changes are required[31,41,48,49], and the controversy will probably only be resolved when the intermediates generated by turnover of a sample in which deactive-like states are not already present are observed. The presence of deactive-like/open states in both the pre-turnover and turnover samples characterised by Kampjut and Sazanov[20] is consistent with both models, as they may represent off-cycle

resting states that are not actively catalysing. A very different mechanism, involving only active-like states, was recently proposed by Gu et al.[25], with an invariably bound $Q_{10}$ molecule shuttling up and down within the Q-binding channel, collecting electrons from N2 and transferring them to a secondary $Q_{10}$ bound in the membrane outside the channel[25]. However, our data suggest the existence of active-state enzyme molecules without $Q_{10}$ bound (active-apo state), and we have not identified any densities from $Q_{10}$ bound outside the channel. Therefore, our data do not align with the proposed 'two-Q' model[25]. Nevertheless, it should be noted that our complex I was not frozen during turnover, and so further evidence is required to confirm the relevance of 'empty' Q-site species to the catalytic cycle.

Common to both interpretations of deactive/open states is the concept that substrate binding triggers restructuring of the Q-binding channel, either during catalysis or reactivation. Here, comparison of our deactive-ligand and deactive-apo structures shows how $Q_{10}$/DDM-binding in the lower section of the channel restructures the NDUFS2-β1-β2 and ND1-TMH5-6 loops in the oxidised deactive state. These restructured loop conformations match the conformations observed in the NADH-reduced open states of ovine complex I, with ubiquinone/ol modelled in a position overlapping with that occupied here by $Q_{10}$/DDM[20]. On the contrary, loop reordering was not observed in porcine complex I in the deactive state[25], in which ubiquinone/ol is modelled in the same position as we have modelled cholate in the Q-binding site in state 3. It is possible that a heterogeneous population of the Q-binding site in the porcine enzyme has led to discontinuous or weaker densities that were not successfully classified into ligand/substrate-bound and -apo states. Finally, in our partially restructured deactive-ligand state the NDUFS2-β1-β2 loop blocks substrate access to the reactive centre, suggesting that the deactive state is resistant to reactivation under the conditions for reverse electron transport (ubiquinol and NAD$^+$)[7] because ubiquinol is unable to react with the oxidised enzyme. As a result, the deactive resting state of complex I protects against the damage caused by reverse electron transfer and the coupled generation of reactive oxygen species in ischaemia-reperfusion injury[13,16,17].

Further modifications of our CxI-ND reconstitution strategy now provide new opportunities to capture and characterise hitherto-unknown catalytic intermediates that are formed as the native $Q_{10}$ substrate binds and is reduced, triggering energy transfer to the proton-pumping membrane domain.

## Methods

**Transformation and recombinant expression of membrane scaffold protein, MSP2N2**. Bacterial strain *Escherichia coli* NiCo21(DE3) was kindly provided by Dr Ali Ryan, Northumbria University, UK. Chemically competent *E. coli* NiCo21(DE3) cells were prepared and transformed with pMSP2N2 (Addgene) using a standard New England BioLabs (NEB) heat shock protocol. For heterologous overexpression of MSP2N2, a starter culture of *E. coli* NiCo21(DE3) containing pMSP2N2 was grown overnight at 37 °C in LB broth with 50 μg mL$^{-1}$ kanamycin. 4 ×2 L Fernbach flasks containing 500 mL of fresh LB media supplemented with antibiotic were inoculated with 1% v/v starter culture and grown at 37 °C, 250 rpm until the culture reached an OD$_{600}$ of 0.6. Protein expression was induced using 0.1 mM IPTG and the cells were cultured for a further 4 h at 37 °C with 250 rpm shaking. Cells were harvested by centrifugation at 6,000 × *g* for 20 min at 4 °C, resuspended in 40 mL lysis buffer (50 mM Tris-HCl pH 8, 500 mM NaCl, 5% glycerol, 1% v/v Triton X-100, 0.002% v/v PMSF, and 1 EDTA-free protease inhibitor tablet (Roche)) and stored at −80 °C.

**Purification of MSP2N2**. The method for purification of MSP2N2 was adapted from a published protocol[34]. Cell suspensions were thawed and lysed by sonication on ice using a Q700 sonicator (Qsonica) (50% output amplitude, 30 cycles of 10 s on, 20 s off). The cell lysate was clarified by centrifugation using a SS34 rotor at 30,000 × *g* for 1 h at 4 °C. The supernatant was collected, syringe filtered through a 0.22 μm membrane (Merck Millipore Ltd.) and applied to a 1 mL Ni-NTA column (His-Trap$^{TM}$ HP, Cytiva) equilibrated with 50 mM Tris-HCl (pH 8), 500 mM NaCl (=buffer A) + 1% v/v Triton X-100. The column was washed with 10 column

volumes of buffer A + 1% v/v Triton X-100 followed by 10 column volumes of buffer A + 50 mM sodium cholate. Non-specifically bound proteins were washed with 10 column volumes of buffer A + 80 mM imidazole and finally MSP2N2 was eluted with buffer A + 400 mM imidazole. To obtain highly pure MSP2N2, the eluate from 1 mL Ni-NTA column was reapplied to a 5 mL Ni-NTA column (His-Trap™ HP, Cytiva) and the same procedure was repeated. Pure MSP2N2 fractions were pooled and dialysed against 2 L of 10 mM MOPS (pH 7.5 at 4 °C), 50 mM KCl at 4 °C. Sample homogeneity was confirmed by SDS-PAGE, and the protein flash frozen and stored at −80 °C.

**Preparation of bovine mitochondria, membranes, and complex I.** Bovine heart mitochondria were prepared as described previously[33], and mitochondrial membranes prepared using a method modified from that used previously for *Mus musculus*[10]. Briefly, mitochondria were thawed and diluted to 5 mg mL$^{-1}$ with 20 mM Tris-HCl (pH 7.55 at 20 °C), 1 mM EDTA, 10% glycerol, then ruptured by three 5 s bursts of sonication with 30 s intervals on ice using a Q700 micro-tip Sonicator (Qsonica) at 65% output amplitude setting. The membranes were pelleted at 75,000 × g using an MLA80 rotor (Beckman Coulter) for 1 h, then resuspended in the same buffer. Bovine complex I was prepared as described previously[33] with a minor modification to match the mouse complex I preparation[10] (solubilised membranes were centrifuged at 48,000 × g for 30 min instead of 8500 × g for 12 min) and kept on ice until reconstitution into nanodiscs.

**Complex I reconstitution into nanodiscs.** Complex I was reconstituted into nanodiscs using a protocol based on the reconstitution of complex I into proteoliposomes[21,35]. Two batches of 0.5 mg of chloroform-dissolved synthetic lipids (1,2-dioleoyl-sn-glycero-3-phosphocholine (DOPC), 1,2-dioleoyl-*sn*-glycero-3-phosphoethanol-amine (DOPE), 18:1 cardiolipin, Avanti Polar Lipids; stock: at a mass ratio of 8:1:1 (DOPC:DOPE:cardiolipin) and total concentration of 25 mg mL$^{-1}$) were each mixed with 20 nmol of chloroform-dissolved ubiquinone-10 (Q$_{10}$; i.e. 40 nmol Q$_{10}$ per mg lipid) in a test tube. The solvent was evaporated off under a stream of N$_2$, and any residual chloroform removed in a desiccator under vacuum for at least 2 h. The dried lipid-Q$_{10}$ mixtures were each resuspended in 457.5 μL 10 mM MOPS (pH 7.5 at 4 °C), 50 mM KCl by vigorous vortexing, and sonicated in an ultrasonic bath (Grant Instruments (Cambridge) Ltd.) for 10 min together with 42.5 μL of 20% w/v solution of sodium cholate (i.e. final concentration of 40 mM). Each sample was transferred into a 1.5 mL Eppendorf tube, centrifuged at 7,000 × g for 10 min in a bench-top centrifuge, then each supernatant was transferred to a new tube and incubated on ice for 10 mins. MSP2N2 and bovine complex I (prepared as described above) were pooled gently with the lipid-Q$_{10}$ mixture at a molar ratio of 400:10:1 (lipid:MSP2N2:complex I). Each sample was then diluted 2-fold with 0.5 mL 10 mM MOPS (pH 7.5 at 4 °C), 50 mM KCl to a total volume of 1 mL, and incubated on ice for 20 min. Then each sample was run over a separate PD10 desalting column (Cytiva) at 4 °C to remove the peripheral detergents. The eluates were pooled together, concentrated using a 100 kDa MWCO Amicon® Ultra concentrator (Merck Milipore Ltd.) to ~100 μL, and filtered using a 0.22 μm Corning® Costar® Spin-X® plastic centrifuge tube filter (Merck Milipore Ltd.). The concentrated sample was applied to a Superose 6 increase 5/150 column (Cytiva) equilibrated with 10 mM MOPS (pH 7.5 at 4 °C), 50 mM KCl, and the most concentrated fractions from the monodisperse CxI-ND peak were used for grid preparation.

**Characterisation of complex I-reconstituted nanodiscs.** The complex I concentration in the nanodisc preparation was quantified relative to a detergent solubilised sample of known concentration using the NADH:APAD$^+$ activity assay with 500 μM APAD$^+$, 1 μM piericidin and 100 μM NADH, as described previously[21,33,35], except that 0.15% soy bean asolectin (Avanti Polar Lipids) and 0.15% CHAPS (Merck Chemicals Ltd.) were present. CxI-ND concentrations (i.e. combined protein concentrations of complex I and MSP2N2) were determined using the Pierce™ bicinchoninic acid (BCA) protein assay kit (Thermo Fisher Scientific). For the sample subjected to cryo-EM analyses, the complex I and CxI-ND concentrations were 3.7 and 4.8 mg mL$^{-1}$, respectively, and the same ratio was observed consistently in several independent preparations. Phospholipid contents were determined as described previously[21,33,35]. The average phospholipid molecular weight was taken to be 771.6 g mol$^{-1}$, and the volume of the phospholipid phase was estimated by assuming that 1 mg of phospholipid occupies ~1 μL. Q$_{10}$ contents were quantified by HPLC, by reference to a set of standard samples, using a Nucleosil 100-5C18 column and a Dionex Ultimate 3000 RS electrochemical detector as described previously[21,33,35]. Q$_{10}$ concentrations were defined relative to the phospholipid phase volume.

All catalytic activity assays were conducted at 32 °C in 96-well plates using a Molecular Devices Spectramax 384 plus plate reader and Softmax Pro. Catalysis was initiated by addition of 200 μM NADH and monitored at 340 and 380 nm ($\varepsilon_{340-380}$ = 4.81 mM$^{-1}$ cm$^{-1}$). Linear rates were used for activity calculations, and inhibitor-insensitive rates (determined by the addition of 1 μM piericidin A) subtracted from each measured rate where noted. Isolated complex I used for reconstitution and CxI-ND samples used for cryo-EM analyses were diluted to 0.5 μg mL$^{-1}$ in 20 mM Tris-HCl (pH 7.5 at 32 °C), and activity assays performed

with 200 μM decylubiquinone (dQ), 0.15% asolectin/CHAPS, and/or 10 μg mL$^{-1}$ alternative oxidase (AOX), prepared as described previously[33].

**Cryo-EM grid preparation and image acquisition.** UltrAuFoil gold grids (0.6/1, Quantifoil Micro Tools GmbH) were prepared as described previously[9]. Briefly, grids were glow discharged (20 mA, 90 s), incubated in a solution of 5 mM 11-mercaptoundecyl hexaethyleneglycol (TH 001—m11.n6-0.01, ProChimia Surfaces) in ethanol for two days in an anaerobic glovebox (Belle), then washed with ethanol and dried just before use. Using a Vitrobot Mark IV (FEI), 2.5 μL of 4.8 mg mL$^{-1}$ CxI-ND solution (from the same preparation) were applied to the grids before blotting for 10 s at force setting −10, at 100% relative humidity and 4 °C, and then plunge-frozen into liquid ethane. Twelve grids were screened for particle number and distribution and two grids were selected for two 2-day data collections. They were imaged using a Gatan K3 detector and a post-column imaging energy filter (Gatan BioContinuum) operating in zero-energy mode with a slit width of 20 eV mounted on an FEI 300 keV Titan Krios microscope (Thermo Fisher Scientific) with a 100 μm and 70 μm objective and C2 apertures, respectively, and EPU v. 2.7.0.5806REL at the Department of Biochemistry, University of Cambridge. Data were collected in super-resolution electron counting mode at a pixel size of 0.535 Å pixel$^{-1}$ (81,000× nominal magnification) with a defocus range −1.0 to −2.4 in 0.2 μm intervals, and the autofocus routine run every 10 μm. Aberration-free image shift (AFIS) was used for data acquisition on day 1 of the 2-day data collection for the first grid but was abandoned due to frequent occurrences of erratic image beam shifts observed in collected movies. The dose rates for the two datasets were 16.9 electrons Å$^{-2}$ s$^{-1}$, with 2.4 s exposures captured in 40 frames. The total dose was thus 40.5 electrons Å$^{-2}$ in both cases. Data were retrieved as non-gain-corrected LZW-compressed tiff movie stacks.

**Cryo-EM data processing.** The two datasets were processed separately until stated otherwise, using RELION 3.1.0[36] (Supplementary Figs. 2 and 3). The micrographs were motion-corrected using RELION's implementation of motion correction with 5 ×5 patches, and contrast transfer function (CTF) estimated using CTFFIND-4.1[50] with an amplitude contrast of 0.1 and ResMax set to 5 Å. Micrographs were filtered to remove those with a negative rlnCtfFigureOfMerit value, an rlnMaxResolution value worse than 6 Å, or an rlnCtfAstigmatism value lower than 20 or greater than 1000. Ice-contaminated micrographs were further removed manually to give 2,639 and 1,797 micrographs for datasets 1 and 2, respectively, from which 804,367 and 382,037 particles were selected using RELION's AutoPicking tool with a 3D map input (EMD-14127)[40] low-pass filtered to 20 Å. Particles were extracted with an initial 4.5× downscaling to 2.4075 Å pixel$^{-1}$, and filtered to select those with an rlnAutopickFigureOfMerit value between 0 and 4. Following one round of 2D (with alignment) and 3D (without alignment) classification to remove junk particles, the remaining particles were re-extracted at the nominal pixel size (1.07 Å pixel$^{-1}$; 2× bin) for another round of 3D classification with local angular search to remove aberrant classes of particles. 251,045 (dataset 1) and 140,731 (dataset 2) particles were brought forward for iterative rounds of CTF refinement[36], to estimate anisotropic magnification, beam tilt, trefoil, 4$^{th}$ order aberration, and per-particle defocus, astigmatism and B-factor parameters. Particles with an rlnNrOfSignificantSamples value greater than 2999 were removed, and the two datasets combined to give 367,615 particles. These particles were then re-extracted with re-centring, subjected to Bayesian polishing, CTF refined and 3D classified (local angular search) to remove any remaining junk. At this early stage, 358,326 particles were 3D refined with a complex I mask (generated from a working model using RELION MaskCreate) and with solvent flattening to give a global resolution of 2.28 Å (according to a gold-standard Fourier shell correlation (FSC) of 0.143). Signal subtraction was then performed to remove most of the non-complex I contribution (MSP2N2 nanodisc belt and the lipid bilayer within the nanodisc) using the complex I mask. 3D classification (number of classes, K = 6, local angular search to 0.2° sampling) was then performed, which separated the particles into active, deactive and state 3 complex I classes. Two classes arising from atypically shaped nanodisc complexes (class 1) and junk (class 2) were excluded from subsequent rounds of processing. Active (class 4), deactive (classes 5 and 6) and state 3 (class 3) classes retained 61,654, 259,547, and 22,019 particles, respectively, which were then signal reverted to include the nanodisc densities, and repolished at 0.7523 Å pixel$^{-1}$ (1.4× bin). The three classes refined to 2.65, 2.28, and 3.02 Å resolution, respectively, at the calibrated pixel size of 0.7496 Å pixel$^{-1}$ (Supplementary Fig. 2) determined by comparison with existing mammalian complex I structures[12]. All data processing were done at the nominal pixel size of 0.7523 Å pixel$^{-1}$, and corrected to a calibrated pixel size of 0.7496 Å pixel$^{-1}$ at the post-processing or local resolution stages in RELION (see below). Focused 3D classification without alignment (regularisation parameter, T = 100) was performed on the active, deactive, and state 3 classes (Table 1 and Supplementary Fig. 3). All individual classes were first subject to signal subtraction to retain roughly only the peripheral arm of complex I, and then focus-classified using a mask generated from a tentatively modelled Q$_{10}$ (active), a mask generated from a provisional partial protein model (ND1, NDUFS2, and NDUFS7) encapsulating the Q-binding site (deactive), or a mask generated from a tentatively modelled DDM (state 3) (Supplementary Fig. 3); 7 junk particles were discarded in this step for the deactive class. Masks for focused classification were generated using RELION MaskCreate with up to 10 pixels of extensions and soft cosine edges; no low-pass filtering was

performed. The identified sub-states outlined in the main text were then signal reverted to give the global map, and the global resolution estimated from the FSC between two independent, unfiltered half-maps (FSC = 0.143) (Table 1 and Supplementary Figs. 3 and 4). As no obvious differences were identified between the state 3 sub-states, which were evenly balanced, they were kept as a single class. The model-generated mask used for 3D refinement with solvent flattening and resolution estimation was generated in UCSF ChimeraX[51] using the *molmap* function, before being low-pass filtered to 15 Å and having a 6-pixel soft cosine edge added using RELION MaskCreate. Mollweide projections were plotted using Python and Matplotlib, and the degree of directional resolution anisotropy calculated using the 3DFSC program suite[52] (Supplementary Fig. 4).

All consensus maps were locally sharpened from the unsharpened, unfiltered half-maps generated from RELION post-process (user-provided B-factor and ad-hoc low-pass filter set to 0 and Nyquist, respectively; the output pixel size was altered to match the calibrated pixel size) using phenix.autosharpen in Phenix 1.18.2-3874[53], setting the resolution limit to the highest local resolution determined from RELION LocalRes (Supplementary Fig. 4), and with a local sharpening box size of 15³ pixels and a targeted overlap of 5 pixels. The deactive-ligand map was split into three sections (distal and proximal membrane domains, and hydrophilic domain) for manual multibody refinement (i.e. signal subtraction, followed by focused refinements) following nanodisc subtraction (Supplementary Fig. 3). The focus-refined maps were then globally sharpened in RELION post-process, and using the globally sharpened consensus deactive-ligand map as a reference, combined to make a composite map using phenix.combine_focused_maps in Phenix 1.19-4092[53] (Supplementary Fig. 3). The composite map was carefully compared to the consensus map to ensure that there were no map distortions or anomalies. The locally sharpened global maps (active-Q₁₀, active-apo, deactive-apo, and state 3) and globally sharpened composite map (deactive-ligand) were used for model building and refinement (Table 1). Nanodisc maps were generated by complex I subtraction and focused refinement (i.e. subtract-refinement) using a nanodisc mask, either with alignment (all particles combined) or without alignment (individual sub-states). To make the nanodisc mask, a tight complex I mask was first generated from a working model using RELION MaskCreate, and subtracted from the consensus map to obtain densities for the nanodisc; leftover complex I densities were removed using the *Map Eraser* tool in UCSF ChimeraX[51], and the complex I-subtracted map then used as the RELION MaskCreate input. Maps were visualised in UCSF ChimeraX[51] for the generation of figures, and raw threshold levels of the relevant maps adjusted using the *Volume Viewer* tool.

**Model building, refinement and validation.** Working models derived from a model for bovine complex I in the active state (PDB ID: 7QSD)[40] were rigid-body fitted into maps using the *Fit in Map* tool in UCSF ChimeraX[51], rigid-body real space refinement in Phenix 1.18.2-3874[53], and Curlew all-atom-refined using Coot 0.9.4-pre[54]. The models were checked, and new resolvable regions built manually in Coot 0.9.4.2-pre. Local bovine populations are known to have a polymorphism at residue position 255 of subunit NDUFA10 – cDNA sequencing has shown evidence for both asparagine and lysine, while electrospray ionisation mass spectrometry supports the latter[55]. On the basis of these reports and the Coulomb potential densities in the CxI-ND maps, we modelled it as Lys255^NDUFA10. Similarly, residue position 129 (glutamine) of subunit NDUFS2 was modelled as Arg129^NDUFS2. Densities for existing and additional phospholipid molecules were identified with the *Unmodelled blobs* tool in Coot. All non-cardiolipin phospholipids were modelled as phosphatidylethanolamines unless density features indicated phosphatidylcholine to be more likely. All DDM molecules in the starting model were removed or replaced with phospholipid molecules where the CxI-ND map features indicated, and lipid tails were clipped where necessary using the delete tools in Coot and PyMOL 2.5.2. The manually inspected models were then real-space refined against the respective locally sharpened consensus (active-Q₁₀, active-apo, deactive-apo, and state 3) or composite (deactive-ligand) maps in Phenix 1.18.2-3874[53] with Ramachandran restraints set to Oldfield (favoured) and Emsley8k (allowed and outlier) to remove genuine forced twists. This real-space refinement step was performed iteratively with manual adjustments in Coot. Water molecules were placed into distinct density peaks as identified with the *Find Waters* function in Coot, with the minimum and maximum distance to protein atoms set to 2.4 and 3.4 Å, respectively. The identified waters were manually edited to remove falsely placed waters (based on H-bonding geometries, strength and shape of densities, and steric clashes) and bulk solvent waters, and to add waters missed due to uncertain positions of surrounding sidechains or waters. Atom resolvabilities (Q-scores) in the respective cryo-EM maps were calculated using MapQ[56], and any outliers identified and corrected. The models were then real-space refined in Phenix as outlined above. The model statistics for the five sub-states in active, deactive and state 3 classes (Table 1) were produced by Phenix, MolProbity, and EMRinger. Model-to-map FSC curves were generated using phenix.validation_cryoem. Model-map CC$_{mask}$ values for various substrate/ligand poses were calculated by first extracting the ligand coordinates from the protein models and then running phenix.validation_cryoem against their respective final maps.

A provisional poly-alanine model for the two MSP2N2 nanodisc belts was built and refined into a 7.4 Å subtract-refined nanodisc map made from all three major species of complex I-reconstituted nanodiscs using interactive molecular dynamic simulations in ISOLDE 1.2.2[57] with α-helical secondary structure restraints.

**Comparisons of cryo-EM maps and models.** Map-to-map real-space correlations were performed using the *Fit in Map* function in UCSF ChimeraX[51] (Supplementary Table 1) following low-pass filtering of the relevant maps to the resolution of the lowest resolution map in the set in RELION[36]. RMSD calculations were performed using the *Align* command in PyMOL.

**Identification of hydrogen bonds.** H-bonding contacts within individual CxI-ND models (with hydrogens added using phenix.ready_set and/or phenix.reduce[53]) were identified using the *hbonds* command in UCSF ChimeraX[51], for which the geometric criteria are based on a survey of small-molecule crystal structures[58], and atom types adapted and extended from the program IDATM[59].

**Quinone cavity determination.** The interior surface of the Q-binding channel was predicted using CASTp[45], which computes a protein surface topology from a PDB model. The default 1.4 Å radius probe was used and the results were visualised in PyMOL using the CASTpyMOL 3.1 plugin and by UCSF ChimeraX[51].

**Molecular dynamics simulations.** The cryo-EM structure of active-state complex I from *M. musculus* at 3.1 Å resolution (PDB ID: 6ZR2)[12] was used to build the initial simulation model. Protonation states of sidechains were adjusted to neutral pH, except that His59^NDUFS2 and Asp160^NDUFS2 were modelled initially as di-protonated (HisH⁺) and neutral (AspH), Glu68^ND3, Glu36^NDUFS5, Glu262^ND1 and Glu114^ND4 as neutral (GluH), and His549^NDUFS1 and His42^NDUFB2 as di-protonated. The N-termini of NDUFS7 (Ser34) and the truncated NDUFB6 (Ser66) were modelled as neutral. These alternative protonation states were suggested by the chemical environment of the group and PropKa calculations[60] (with nearby FeS cluster N2 in the oxidised state). All cofactors and post-translational modifications present in PDB 6ZR2 were included. High confidence phospholipids were retained and built with linoleoyl (18:2) acyl chains. The polar headgroups were preserved, thereby they were modelled as 1,2-dilinoleoyl-sn-glycero-3-phosphatidylcholine (DLPC), 1,2-dilinoleoyl-sn-glycero-3-phosphatidylethanolamine (DLPE) and 1'-3'-bis[1,2-dilinoleoyl-sn-glycero-3-phospho]-sn-glycerol, as the cardiolipin dianion (CDL). Missing hydrogen atoms were built with the GROMACS suite[61]. Neutral His tautomers were chosen based on optimal H-bonding.

The modelled protein complex was embedded in a solvated bilayer with a composition mimicking that of the inner mitochondrial membrane[62]. The final solvated and inserted system contains 368 DLPC (179 in the matrix leaflet), 294 DLPE (143 in the matrix leaflet), 96 CDL (half in each leaflet) and 22 oxidised Q₁₀ molecules, all initially in the membrane phase. The asymmetry in DLPC and DLPE composition between the two leaflets is due to chain NDUFA9 occupying an area of the matrix leaflet only. The water phase contains 205,387 molecules plus 553 Na⁺ and 348 Cl⁻ ions to neutralise the total system charge and maintain ~0.1 M salt concentration. The resulting simulation model contains a total of 861,975 atoms. Water solvation of the apo Q-binding channel and regions near the oxidised N2 FeS centre was compared to the active-Q₁₀ and previous cryo-EM models (PDB ID: 6YJ4)[23] and adjusted accordingly.

The simulation model was relaxed and equilibrated during molecular dynamics (MD) simulations of 740 ns in total. Initially all protein heavy atoms were tethered to their initial position by harmonic restraints, then the force constants were decreased progressively from 1000 to 10 kJ mol⁻¹ nm⁻¹, and all atoms were free to move in the last 210 ns. Membrane packing (area per lipid) and hydration of several groups in the Q-binding channel were monitored and checked for stability after 400 ns. Then, the final configuration of this simulation had one Q₁₀ molecule inserted into the binding channel. The Q₁₀ coordinates, as well as those of the sidechains of His59^NDUFS2, Tyr108^NDUFS2, Thr156^NDUFS2, Met70^NDUFS7 and Ser205^ND1, were adjusted to the superimposed active-Q₁₀ cryo-EM model (Fig. 2a). Clashing water molecules in the channel were removed and the protonation states of His59^NDUFS2 and Asp160^NDUFS2 were changed to the three charge-states studied ([Asp⁻ + His]; [AspH + His]; [AspH + HisH⁺]). If necessary, charge neutrality was maintained by removing a counter-ion. Each charge-state was further relaxed and equilibrated during 235 ns of MD simulation, with initial harmonic restraints in heavy protein atoms, the distances between atoms His59^NDUFS2-N$_{\delta 1}$–Q₁₀O₃ and His59^NDUFS2-N$_{\epsilon 2}$–Asp160^NDUFS2-C$_\gamma$, and the collective variable (CV) described below. Restraint forces were progressively reduced to zero and removed fully in the last 50 ns. Hydration of the Q-binding channel was again monitored and checked for stability. This procedure was designed and implemented to generate equilibrated and unbiased simulation models in the three charge-states studied. A canonical MD trajectory of 300 ns without any restraints was obtained for each charge-state. The root-mean squared deviation (RMSD) for the positions of the C$_\alpha$ atoms of chains NDUFS7, NDUFS2 and ND1 remained stable during these trajectories at ~1.4 Å in relation to both the initial model (PDB 6ZR2) and the current active-Q₁₀ cryo-EM model.

A pathway collective variable (CV), as used previously for Q-binding simulations[31], was applied to describe the position of the Q-headgroup along the channel. This CV (see Fig. 3b–d) is a combination of distances between the heavy atoms in the Q-headgroup and the C$_\alpha$ atoms of residues in subunits NDUFS7, NDUFS2 and ND1 exposed to the Q-binding channel. Distances are evaluated with respect to four milestone configurations that represent progressive binding of Q₁₀.

These configurations are provided here as Supplementary Data 1 to allow reproduction of our calculations.

Finally, well-tempered metadynamics[63] simulations were performed for each charge-state, starting from a configuration taken at 75 ns of each canonical MD trajectory described above. Metadynamics were activated in the CV coordinate (position on the path, p1.sss, and distance from the path, p1.zzz) and in the His59$^{NDUFS2}$ $\chi_2$ dihedral (C$_\beta$-C$_\gamma$ bond torsion) with Gaussians deposited every 500 time steps (1 ps), at initial height of 0.6 kJ mol$^{-1}$, widths of 0.4 and 0.02 units for the CV and dihedral, respectively, and a bias factor of 15.0. Walls were included to restrict sampling for the CV at 21 < p1.sss < 26 Å and −2.45 < p1.zzz < −2.20 Å, with force constant of 1000 kJ mol$^{-1}$ nm$^{-1}$. Productive metadynamics simulations lasted 140 ns for each charge-state. Due to the enhanced sampling nature of this method, simulation times significantly shorter than for canonical MD are sufficient for an appropriate conformational sampling of confined regions, such as the reactive position of the complex I Q-binding site. Convergence within ±1 kJ mol$^{-1}$ of free energy differences in the CV profile (Fig. 3a) was reached after 70 ns. The effects of metadynamics and of restraints were removed by re-weighting the distribution of structural properties (distances, dihedral, CV) and the resulting free energies are shown in Fig. 3a, e–g and Supplementary Fig. 8. The statistical uncertainty was estimated as 95% confidence intervals by bootstrap analysis.

In all MD simulations the interactions of protein, lipids and ions were described with the all-atom CHARMM36m force-field[64]. Water was represented by the standard TIP3P model[65]. FeS centres were described using the Chang and Kim[66] parameters with corrections by McCullagh and Voth[67]. Q$_{10}$ interactions were represented by our calibrated force-field[62,68]. The remaining cofactors were described by available CHARMM and CGenFF parameters (charmm36-mar2019.ff)[64]. All simulations were conducted with GROMACS (version 2020.3)[61] at constant temperature of 310 K, pressure of 1 atm and a time step of 2 fs. Long-range electrostatics were treated with the Particle Mesh Ewald method[69]. Metadynamics simulations were performed with the PLUMED plugin (version 2.6.1)[70].

**Reporting summary.** Further information on research design is available in the Nature Research Reporting Summary linked to this article.

## Data availability

The data that support the findings of this study are available from the corresponding author upon reasonable request. Structural data have been deposited in the EMDB and PDB databases under the following accession codes: EMD-14132 and 7QSK (active-Q$_{10}$), EMD-14133 and 7QSL (active-apo), EMD-14134 and 7QSM (deactive-ligand; composite), EMD-14135, EMD-14136, [EMD-14137], and EMD-14138 (deactive-ligand; consensus, hydrophilic domain, proximal and distal membrane domains, respectively), EMD-14139 and 7QSN (deactive-apo), and EMD-14140 and PDB 7QSO (state 3). Related data accession codes: EMD-14127 and 7QSD (DDM-solubilised bovine complex I).

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

## Acknowledgements

We thank D. Chirgadze, S. W. Hardwick and L. Cooper (University of Cambridge Cryo-EM facility) for assistance with grid screening and cryo-EM data collection during the COVID-19 pandemic with restricted access; A. Ryan (Northumbria University, UK) for the *E. coli* NiCo21(DE3) expression strain for MSP2N2; the SDumont cluster in the National Laboratory for Scientific Computing (LNCC/MCTI, Brazil) for computational resources; and D. N. Grba (MRC MBU) for critical feedback. This work was supported by the Medical Research Council (MC_UU_00015/2 to J.H.), the Swiss National Science Foundation (P400PB_191096 to O.B.) and Fundação de Amparo à Pesquisa do Estado de São Paulo (FAPESP, grant 2019/21856-7 to G.M.A. and fellowship 2020/14542-3 to C.S.P.).

## Author contributions

I.C. performed CxI-ND reconstitution, cryo-EM grid preparation, collected and processed cryo-EM data, carried out structure model building, analyses, and interpretations, and prepared figures. I.C. and J.J.W. carried out biochemical and kinetic characterisations. J.J.W. prepared complex I for reconstitution, assisted by I.C. H.R.B. advised on cryo-EM data collection and processing and built initial models. J.J.W. and H.R.B. contributed to data interpretation. B.S.I. prepared MSP2N2 for reconstitution, with contributions from J.J.W. O.B. established and optimised the initial reconstitution protocol. C.S.P. and G.M.A. performed and analysed molecular simulations. J.H. initiated and supervised the project and contributed to interpretation of the data. I.C. and J.H. wrote the paper with input from all authors.

## Competing interests

The authors declare no competing interests.
