## [Peer Review File · Nature Communications]

Cryo-EM structures define ubiquinone-10 binding to mitochondrial complex I and conformational transitions accompanying Q-site occupancyReviewers' Comments:

Reviewer #1:

Remarks to the Author:

This manuscript by Chung et al. presents a new cryo-EM structure of mitochondrial complex I reconstituted into phospholipid nanodiscs accompanied with molecular dynamics simulations to analyze the quinone-binding site of the enzyme in relation to active/deactive (A/D) transition. This is a very well-written manuscript that presents a new set of data and brings original insights on the mechanism of complex I functioning.

The work is sound and the manuscript is high quality. This reviewer has only a few questions and minor comments.

1. Introduction, 1st paragraph

The existence of "Two biochemically characterised, physiologically relevant states, the 'active' and 'deactive' resting states" have been initially proposed and analyzed by Vinogradov's group in the 90's (Refs 14-15) before advances in Cryo-EM/crystallography studies of complex I structure. The beginning of that paragraph should be rephrased to give a reader an unbiased introduction to this subject.

2. Page 5, 1st Results Section and SupplFig. 1

Short chain homologue decyl ubiquinone was used as a substrate for activity measurements, and NADH:dQ reductase could be observed only in the presence of asolectin/CHAPS. At the same time, authors reconstitute complex I to nanodiscs in the presence of natural Q10. What would happen if dQ is used for the reconstitution instead?

3. In the Materials and Methods section the authors mentioned that

"Linear rates were used for activity calculations, and inhibitor-insensitive rates (determined by the addition of 1 μ M piericidin A) subtracted from each measured rate where noted."

while in the Suppl Fig. 1B legend

"Note that inhibitor-insensitive background rates have not been subtracted."

To clarify this, activities with and without piericidin can be presented in Suppl. Fig. 1B.

4. Page 18 First paragraph

The proposal of Gu et al., 2022 regarding the existence of a second Q-binding site at which bulk membrane Q10 accepts electrons from a "bound" Q10H2 within complex I is worth a more detailed discussion. How is it related to the position of the Q10 molecule reported in the present manuscript?

5. Page 19 1st paragraph

The authors stated that "in deactive-ligand state the NDUF52- β 1- β 2 loop blocks substrate access" and it "suggests" that the D-form is unable to catalyze reverse electron transfer. In fact, experiments showing the inability of the deactive enzyme to catalyze this reverse reaction have been performed and reported in the original paper describing the A/D transition in the 90s' (Ref 14 of the present manuscript). This sentence could be amended to give a correct view on this subject.

Minor comments

1. Page 16 Line 8

What is "local protonation equilibria" in the relation to protonation/deprotonation of the functional group?

2. There is a typo in Supplemental Materials:

"compatent" – should be amended

Reviewer #2:

Remarks to the Author:

In their manuscript "Cryo-EM structures define ubiquinone-10 binding to mitochondrial complex I and conformational transitions accompanying Q-site occupancy" Chung et al. describe the structures of bovine (*Bos taurus*) mitochondrial complex I reconstituted into nanodiscs in the presence of the substrate ubiquinone-10 (Q10). The authors identify 5 distinct states of the complex present in their preparation and two distinct modes of Q10 binding to the active complex. They also identify an apo

state of the active complex that lacks Q10, which is important as this state was proposed to not be present from a recent structural study of porcine respiratory supercomplexes (PMID: 35145322). From molecular dynamics simulations they conclude that the mode of Q10 binding observed in their structures represents a 'pre-reactive' state. Overall, the paper presents important new results on Q10 binding to mammalian mitochondrial CI, clarifies discussions around the observed distinct states of the complex and presents a new biochemical system in which to further study the mechanism of energy-conversion by the complex. The manuscript is clearly written and will be of great interest to people in the field.

Comments

Given the specific features of the nanodisc adjacent to the entry site of the Q-tunnel in ND1 below NDUFA9 and that conformational changes and lipid binding in this region are important features of the A/D transition as well as specific complex I mutants (PMID: 33243981), the authors should more thoroughly discuss differences in this region between their different states. Specifically, in the context of any observed lipids and the Q10 binding site recently reported in this region (PMID: 35145322). Is the Q10 binding site seen in the porcine structure well-ordered in active state of the bovine complex?

The map contours referred to in the figures should be made more explicit. Do they refer to sigma values of a normalized map? If so, it should be presented as "4.5" sigma (s) throughout. If not, how are the contour levels and map values determined?

Given the need for local refinements of the deactive ligand classes indicated multiple deactive states with distinct angles between the peripheral and membrane arms (as has been observed previously PMID: 31492636) and the large number of particles in the deactive-ligand class (235,957) were additional classification strategies tried to separate distinct "open" classes as has been previously observed (PMID: 32972993). Given that the observed ligand density clearly represents multiple distinct molecular species bound in the Q-tunnel, it would be important to determine whether classifying based on differences in the angle between the peripheral and membrane arm using a focus-revert-classify approach (PMID: 31492636) may reveal distinct states that are more homogeneous with respect to the occupancy of the Q-tunnel.

Minor comments

Line 228-229, incomplete sentence fragment, "...and so further classification of the deactive particles focussed on the whole Q-binding region"needs revision. Plus typo in spelling of focused.

Line 271, "...but in different conformations to in the deactive-ligand or active states..." should be "...but in different conformations to the deactive-ligand or active states..."

Line 333, the sentence ends with "this structure" but it is unclear exactly which structure "this structure" is referring to.

Line 739 Should "DNPC" and "DNPE" be "DLPC" and "DLPE"?

Reviewer #3:

Remarks to the Author:

This paper determines high-resolution cryo-EM structures of the mitochondrial complex I in a lipid nanodisc with exogenous ubiquinone-10 (Q10). The results demonstrate multiple states of the protein assembly complex, providing mechanistic insight into the substrate binding and the pivotal entry step for the electron transport chain (ETC). The authors presented a high-quality study in the

structural biology of complex I. However, some detailed clarifications and revisions will need to be addressed before publication.

Major concerns:

1. The authors described the advantage of studying the complex I reconstituted in a lipid nanodisc, which mimics a native-like environment. However, in this case, the structures did not seem significantly changed, and the function of the complex I that links to the surrounding lipids was not addressed. It is recommended to describe the significance of lipid-protein interactions in the structure and how the synthetic phospholipids for reconstitution were chosen. Also, 20 more lipids were newly resolved in this study, and are the binding locations overlapped with the previous structure? In addition, was the resolution sufficiently high enough, or was the map quality high enough to resolve the lipid head groups? What were the distributions of these lipids? And do they play any role in the function of complex I?
2. It could be helpful to show the chemical or protein components in the analyzed sample, which provides confidence in modeling the molecules in the cryo-EM densities. Mass spectrometry or LC-MS/MS may provide another layer of molecular evidence in modeling unknown densities in this study.

Minor concerns:

1. It is strongly recommended not to present a specified protein residue by superscripting the protein name, such as Tyr108^{NDUFS2}. 'NDUFS2-Tyr108', 'NDUFS2-Y108', or 'Tyr108 of the NDUFS2' is most seen. These changes can dramatically improve readability.
2. INTRODUCTION (paragraph 2): "... Cyro-EM studies of mammalian complex I have also identified a third state of particles that lack key regions of density, which ..."
 "... Cryo-EM studies of mammalian complex I have also identified a density of the third state that lacks the information of the key regions, which ..."
3. INTRODUCTION (paragraph 3): "... As expected, the structure shows a fully-bound Q10 spanning the entirety of the channel, but, consistent with the observations of the short-chain substrate ..."
4. INTRODUCTION (paragraph 4): "... By comparing the structures of substrate/ligand-bound and apo (substrate/ligand-free) forms ..."
5. The author described that the intact nanodisc-reconstituted CxI-NDs displayed little piericidin-sensitive NADH:dQ activity. Would it be possible to analyze whether the reconstituted lipid or previously used detergent plays any role in function? Also, it seems that the active site is away from the lipid bilayer, and how would the bilayer reconstitution interfere with the dQ entering or function?
6. To completely remove the effect from the DDM detergent, has the author tested using BioBeads to ensure the residual detergent is minimal?
7. RESULTS(Resolution of three major classes of CxI-ND particles, paragraph 2): "... As reported previously, the state 3 map (see Supplementary Fig. 5) lacks clear densities for the C-terminal half of the ND5 transverse helix and ..."
8. It is recommended to use colors that are friendly to the color-blinded, especially the colors in Fig. 1.
9. FIGURE 1: It would be appreciated if the author could show the views of the model or the density in orthogonal directions. Ideally, the membrane plane will be perpendicular or parallel to the viewing directions.
10. FIGURE 2 AND 5: Please label the distances for potential hydrogen bonds or electrostatic interactions.
11. FIGURE 2 AND 6: Please describe the green surfaces in the panels in figure legends.
12. FIGURE 4: It is recommended to show the green cavities in a closed form.
13. Like a cholesterol molecule, the cholate molecule has α (smooth) face and β (spiky) faces, featuring its interactions with specific residues. For example, the α face prefers to interact with aromatic residues, whereas the β face favors the residues of Ile, Val, and Leu. When modeling the cholate coordinates into the cryo-EM density, do the authors find a similar pattern of interactions?
14. METHODS(Complex I reconstitution into nanodiscs; paragraph 1): "... and total concentration of 25 mg mL⁻¹ were each mixed with 200 nmol ..."
15. The nominal magnification described in 'Cryo-EM grid preparation and image acquisition' of the Method section did not seem correctly calculated using a 5-micrometer pixel size of the Gatan K3 DED

camera. Also, in the reported Table 1, the magnification was reported as 130,000X (affected by the post-column energy filtering), which did not match the description in the Method section. The author may need to correct one of them to make it consistent.

16. The author will need to clarify further how the pixel size in the density map was calibrated (0.7496 Å/pixel) from the measured (0.535 Å) (Table 1). When the calibrated pixel size was used, was the CTF estimation or refinement re-performed? Was the model density re-refined and re-adjusted to the correct scaling for individual Fourier pixels?

17. METHOD(Cryo-EM data processing; paragraph 1): Please rewrite the second sentence, which was not clearly written. The name of the 'MotionCorr' is referred to another program for motion correction, which should not be written as "RELION's implementation MotionCorr". The amplitude contrast is used in contrast transfer function estimation, which should be separated into an independent sentence from what was described for the motion correction.

18. METHOD(Cryo-EM data processing; paragraph 1): What is the 3D reference used for automated particle picking? Was there any pre-treatment on the 3D reference?

19. METHOD(Cryo-EM data processing; paragraph 1): The sentence mentions the signal subtraction, which was performed to remove most of the non-complex I contribution. What is the non-complex I contribution and where are they?

20. It is recommended not to present individual classes as C1, C2, C6, etc. A general description of 'class #' will be sufficient.

21. It may be a concern to use 'masking' and 'solvent flattening' interchangeably in the Method section, although the purposes of the two processing procedures are similar.

22. It seemed that the ligand-bound state (2.30 Å resolution, 235,957 particles) has the largest particle population among all the others. Does it provide any implication in the energy landscape of the complex? Would it be possible to project the state onto the MD trajectory?

23. It is recommended to show the quality of the model fitting, especially the densities relevant to the function, in Supplementary Data.

24. The authors modeled the atomic coordinates of the water molecules in multiple states. Are the coordinates of the waters consistent across these different structures? How are the water distributions in the bilayer with protein and outside the nanodisc? It would be helpful to look at the models along with their cryo-EM densities.

25. It is recommended to add 3DFSC plots for individual structures to show whether the resolutions were determined and measured isotropically or anisotropically.

26. SUPPLEMENTARY FIG. 2: It is recommended to show the scale of the 2D class averages.

27. SUPPLEMENTARY FIG. 4: It is recommended to show red (low) to blue (high) for local resolutions.

Reviewer #4:

Remarks to the Author:

The manuscript by Chung et al. reports a cryo-EM study of a mammalian complex I enzyme reconstituted into phospholipid nanodiscs with exogenous Q10 combined with enhanced sampling molecular dynamics simulations. Five distinct cryo-EM structures in different functional states are resolved at 2.3-3.0 Å resolution including one with a Q10 molecule occupying the full length of the Q-binding channel. These structures combined with molecular dynamics based free energy calculations provide a mechanistic picture on how substrate binding restructures the deactive Q-binding site.

In my opinion, this is an interesting study and has some important mechanistic implications in terms of Complex I ubiquinone interactions and the role of protonation states of D160 and H59 in these interactions. I only have a few comments that need to be addressed by the authors to clarify and improve the manuscript:

1. As the authors also state both 'active' and 'deactive' resting states of Complex I have previously been characterized by cryo-EM. There is also a recent cry-EM structure of Q10-bound Complex I that the manuscript refers to in several places. It seems to me the distinction the authors try to make

between their work and previously published work (in terms of their conclusions and not methodology) relies more on the MD side of the study. My question is why they only use one of their 5 cryo-EM structures to do MD and more specifically why they use the apo active state rather than the active-Q10 state and insert the Q10 molecule instead. Since MD is really what distinguishes this work from the previous ones, perhaps, they should expand more on this side of their study.

2. A methodological aspect of this work that distinguishes it from Ref. 27, for instance, is the use of nanodiscs. The authors state in the Introduction that "the nanodiscs provide a native membrane-like environment and eliminate potential artifacts from the detergent micelle typically present in cryo-EM analyses". However, in the Results section, there is no evidence of any differences. I think this needs to be clarified. There is a bit of a disconnect between the Introduction and Results.

3. I am somewhat surprised by the long restraining stage in the MD simulations, which is longer than the production run if I read the Methods section correctly. Is there a reason the authors needed such long restraining stage?

4. I think the free energy calculation part needs proper error analysis. The authors state: "Convergence within ± 1 kJ/mol of free energy differences in the CV profile was reached after 70 ns". However, the apparent convergence does not necessarily give a good sense of error. A common technique used in the free energy calculation field is bootstrapping that can provide better estimates of the uncertainty.

Reviewer #1

This manuscript by Chung et al. presents a new cryo-EM structure of mitochondrial complex I reconstituted into phospholipid nanodiscs accompanied with molecular dynamics simulations to analyze the quinone-binding site of the enzyme in relation to active/deactive (A/D) transition. This is a very well-written manuscript that presents a new set of data and brings original insights on the mechanism of complex I functioning. The work is sound and the manuscript is high quality. This reviewer has only a few questions and minor comments.

We thank the reviewer for this positive evaluation.

1. Introduction, 1st paragraph

The existence of “Two biochemically characterised, physiologically relevant states, the ‘active’ and ‘deactive’ resting states” have been initially proposed and analyzed by Vinogradov’s group in the 90’s (Refs 14-15) before advances in Cryo-EM/crystallography studies of complex I structure. The beginning of that paragraph should be rephrased to give a reader an unbiased introduction to this subject.

The reviewer is correct and we apologise this oversight. We have amended our text accordingly: *Two biochemically characterised, physiologically relevant states, the ‘active’ and ‘deactive’ resting states described initially by Vinogradov and coworkers (ref. 7,8), have previously been identified by electron cryomicroscopy (cryo-EM) on mammalian complex I (ref. 3,9,10).*

2. Page 5, 1st Results Section and SupplFig. 1

Short chain homologue decyl ubiquinone was used as a substrate for activity measurements, and NADH:dQ reductase could be observed only in the presence of asolectin/CHAPS. At the same time, authors reconstitute complex I to nanodiscs in the presence of natural Q10. What would happen if dQ is used for the reconstitution instead?

We thank the reviewer for raising this point. dQ is able to partition from the aqueous phase into the phospholipid phase following reconstitution (unlike Q₁₀, which has a higher logP so cannot be added to the aqueous phase). There is thus no advantage to adding dQ to the reconstitution. Furthermore, dQ added to a phospholipid phase readily partitions back out into the aqueous phase – and so it would not be retained in the nanodiscs following reconstitution. Because of the rapid equilibration of dQ between phases it is necessary to add it when it is needed – for example in the assays shown in Supplementary Fig. 1b.

3. In the Materials and Methods section the authors mentioned that

“Linear rates were used for activity calculations, and inhibitor-insensitive rates (determined by the addition of 1 μM piericidin A) subtracted from each measured rate where noted.” while in the Suppl Fig. 1B legend “Note that inhibitor-insensitive background rates have not been subtracted.” To clarify this, activities with and without piericidin can be presented in Suppl. Fig. 1B.

We apologise for the lack of clarity. Unfortunately, it was not possible to measure every inhibitor-insensitive background rate for the data shown in Supplementary Fig. 1b as these data were recorded using the exact same sample as used for cryo-EM analysis, and there was insufficient sample remaining after cryo-EM grid preparation and other biochemical characterisations. We have been careful to denote the values reported in the main text as piericidin-sensitive values, however we lack the data to report piericidin-sensitive data throughout the comparison in Supplementary Fig. 1b. We have now provided additional clarification in the figure legend: *The data reported were recorded on the sample analysed by cryo-EM; note that inhibitor-insensitive background rates were not recorded in some cases due to lack of sample availability, and therefore all the data in the figure are reported without background subtraction to enable their comparison.*

4. Page 18 First paragraph

The proposal of Gu et al., 2022 regarding the existence of a second Q-binding site at which bulk membrane

Q10 accepts electrons from a “bound” Q10H2 within complex I is worth a more detailed discussion. How is it related to the position of the Q10 molecule reported in the present manuscript?

We thank the reviewer for raising this important point. The proposal of Gu et al., 2022 relies on an invariably bound Q₁₀ molecule (assigned in all their structures) that shuttles up and down the binding channel. However, as noted in our manuscript, we observe a Q₁₀-free active state (active-apo state) that fails to align with this model. Additionally, we do not observe a secondary Q₁₀ bound at the entrance of the Q-binding site in our active state to accept electrons from the channel-bound Q₁₀. However, it must be noted that our complex I sample was not frozen ‘during turnover’ so the states we observe may be ‘off-cycle’ states, not representative of catalytic intermediates. It is therefore not possible to argue against the ‘two-Q’ model with certainty. We have expanded our text as follows: *However, our data suggest the existence of active-state enzyme molecules without Q₁₀ bound (active-apo state), and we have not identified any densities from Q₁₀ bound outside the channel. Therefore, our data do not align with the proposed ‘two-Q’ model (ref. 25). Nevertheless, it should be noted that our complex I was not frozen during turnover, and so further evidence is required to confirm the relevance of ‘empty’ Q-site species to the catalytic cycle.*

5. Page 19 1st paragraph

The authors stated that “in deactive-ligand state the NDUFS2-β1-β2 loop blocks substrate access” and it “suggests” that the D-form is unable to catalyze reverse electron transfer. In fact, experiments showing the inability of the deactive enzyme to catalyze this reverse reaction have been performed and reported in the original paper describing the A/D transition in the 90s’ (Ref 14 of the present manuscript). This sentence could be amended to give a correct view on this subject.

We thank the reviewer for bringing this earlier work to our attention and have now added ref. 14 (now ref. 7) to this sentence. However, we note that we are suggesting only an explanation for the fact that D is unable to catalyse in reverse: *the NDUFS2-β1-β2 loop blocks substrate access to the reactive centre, suggesting that the deactive state is resistant to reactivation under the conditions for reverse electron transport (ubiquinol and NAD+) (ref. 7) because ubiquinol is unable to react with the oxidised enzyme.*

Minor comments

1. Page 16 Line 8

What is “local protonation equilibria” in the relation to protonation/deprotonation of the functional group?

We apologise if this was not clear, we have clarified our text to read: *shifting ionisation equilibria to protonate His59^{NDUFS2}.*

2. There is a typo in Supplemental Materials:

“compatent” – should be amended

Thank you, the spelling has now been corrected.

Reviewer #2

In their manuscript “Cryo-EM structures define ubiquinone-10 binding to mitochondrial complex I and conformational transitions accompanying Q-site occupancy” Chung et al. describe the structures of bovine (*Bos taurus*) mitochondrial complex I reconstituted into nanodiscs in the presence of the substrate ubiquinone-10 (Q10). The authors identify 5 distinct states of the complex present in their preparation and two distinct modes of Q10 binding to the active complex. They also identify an apo state of the active complex that lacks Q10, which is important as this state was proposed to not be present from a recent structural study of porcine respiratory supercomplexes (PMID: 35145322). From molecular dynamics simulations they conclude that the mode of Q10 binding observed in their structures represents a ‘pre-reactive’ state. Overall, the paper presents important new results on Q10 binding to mammalian mitochondrial CI, clarifies discussions around the observed distinct states of the complex and presents a

new biochemical system in which to further study the mechanism of energy-conversion by the complex. The manuscript is clearly written and will be of great interest to people in the field.

We thank the reviewer for this positive evaluation.

Comments

Given the specific features of the nanodisc adjacent to the entry site of the Q-tunnel in ND1 below NDUFA9 and that conformational changes and lipid binding in this region are important features of the A/D transition as well as specific complex I mutants (PMID: 33243981), the authors should more thoroughly discuss differences in this region between their different states. Specifically, in the context of any observed lipids and the Q10 binding site recently reported in this region (PMID: 35145322). Is the Q10 binding site seen in the porcine structure well-ordered in active state of the bovine complex?

i) The PDB model files and EMD maps for the recent structural study of porcine respiratory supercomplexes (PMID: 35145322) were not publicly accessible when we submitted our manuscript, so it was not possible to carry out any detailed comparisons then. We have now examined the Q₁₀ binding site region in subunit NDUFA9 of the active state porcine complex in our active-Q₁₀ and active-apo models and maps. The region is well ordered and a phospholipid (modelled as a phosphatidylethanolamine) is observed in place of the Q₁₀ molecule modelled in the porcine enzyme. We have added a sentence on page 14 to explain that: *In our active states, this unorthodox Q₁₀-binding site is well ordered and occupied by a phospholipid molecule.*

ii) We thank the reviewer for highlighting the relevance of NDUFA9, and for prompting us to re-examine this subunit in our models. In comparison to previous detergent-bound structures of deactive/open states, we are able to model substantially more of the C-terminal region of NDUFA9 than has previously been possible. Although the resolution is still relatively low, so that we do not wish to comment on phospholipid/Q₁₀ binding in our deactive states, our results suggest that the difference between the active and deactive NDUFA9 subunits in the membrane may be less extensive than previously supposed. We describe our observations on page 8: *For the deactive-state structures, we were able to model substantially more of subunit NDUFA9 than has previously been possible in detergent-solubilised deactive/open structures (ref. 3,9,10,13,20,25). Only residues 324-331, adjacent to the disordered loop in ND3, were not modelled, suggesting that the nanodisc environment is able to contain the disorder that further propagates to residues ca. 186-195, 253-278 and 323-334 (ref. 13,20,25) in the detergent-solubilised enzyme. It is thus possible that structural changes to NDUFA9 in the membrane-bound deactive state are less extensive than previously supposed.*

The map contours referred to in the figures should be made more explicit. Do they refer to sigma values of a normalized map? If so, it should be presented as “4.5” sigma (s) throughout. If not, how are the contour levels and map values determined?

We apologise for the lack of clarity. The map contour levels are not sigma values. They refer to the raw values from the map file (.mrc or .ccp4) that is used by UCSF ChimeraX for visualisation. This is why we stated in all figure legends that the maps are *presented at a contour level of # in UCSF ChimeraX* to avoid confusion with sigma values. We have now clarified this by using the term *map threshold* instead of *contour level* throughout the text, and we have also added a sentence at the end of the ‘Cryo-EM data processing’ section of the methods on page 39: *Maps were visualised in UCSF ChimeraX (ref. 50) for the generation of figures, and raw threshold levels of the relevant maps adjusted using the Volume Viewer tool.*

Given the need for local refinements of the deactive ligand classes indicated multiple deactive states with distinct angles between the peripheral and membrane arms (as has been observed previously PMID: 31492636) and the large number of particles in the deactive-ligand class (235,957) were additional classification strategies tried to separate distinct “open” classes as has been previously observed (PMID: 32972993). Given that the observed ligand density clearly represents multiple distinct molecular species bound in the Q-tunnel, it would be important to determine whether classifying based on differences in the angle between the peripheral and membrane arm using a focus-revert-classify approach (PMID: 31492636) may reveal distinct states that are more homogeneous with respect to the occupancy of the Q-tunnel.

We thank the reviewer for raising this important point, and indeed we have invested considerable effort in investigating it. As shown in the global classification figure (Supplementary Fig. 2) but not discussed in the manuscript, two classes (C5 and C6) representing the deactive class were initially separated, but later combined. These two classes showed very mild differences in the angle between the peripheral and membrane arms of complex I (as would be observed from a focus-revert-classify approach), but detailed investigation of the maps suggested that there were no fundamental differences between them (including at the Q-binding site) other than the minor difference in inter-arm angle, which is why the two classes were re-combined. We also attempted a tighter focused 3D classification of the ligand density in the deactive-ligand class (mimicking that of the successful focused 3D classification for the active class), but we were unable to successfully classify out the heterogeneity at the entrance of the Q-binding site. We speculate that this is because of the substantial overlap between the DDM and Q₁₀ positions at the entrance of the Q-binding site and the ligand density being too small with respect to the surrounding protein density.

Minor comments

Line 228-229, incomplete sentence fragment, "...and so further classification of the deactive particles focussed on the whole Q-binding region" needs revision. Plus typo in spelling of focused.

Thank you for this comment. We have amended the sentence: *...and so further classification of the deactive particles was focused on the whole Q-binding region. All instances of 'focussed' in the text have been corrected to 'focused'.*

Line 271, "...but in different conformations to in the deactive-ligand or active states..." should be "...but in different conformations to the deactive-ligand or active states..."

Thank you. We have corrected the mistake.

Line 333, the sentence ends with "this structure" but it is unclear exactly which structure "this structure" is referring to.

We apologise for the confusion: "this structure" refers to the dQ-bound *T. thermophilus* structure (ref. 19) mentioned just prior to the phrase in question. We now write *...that increases the separation of Tyr108^{NDUFS2} and His59^{NDUFS2} in this dQ-bound bacterial structure.*

Line 739 Should "DNPC" and "DNPE" be "DLPC" and "DLPE"?

We thank the reviewer for noting this error and have corrected it accordingly.

Reviewer #3

This paper determines high-resolution cryo-EM structures of the mitochondrial complex I in a lipid nanodisc with exogeneous ubiquinone-10 (Q10). The results demonstrate multiple states of the protein assembly complex, providing mechanistic insight into the substrate binding and the pivotal entry step for the electron transport chain (ETC). The authors presented a high-quality study in the structural biology of complex I. However, some detailed clarifications and revisions will need to be addressed before publication.

We thank the reviewer for this positive evaluation.

Major concerns:

1. The authors described the advantage of studying the complex I reconstituted in a lipid nanodisc, which mimics a native-like environment. However, in this case, the structures did not seem significantly changed, and the function of the complex I that links to the surrounding lipids was not addressed. It is recommended to describe the significance of lipid-protein interactions in the structure and how the synthetic phospholipids for reconstitution were chosen. Also, 20 more lipids were newly resolved in this study, and are the binding

locations overlapped with the previous structure? In addition, was the resolution sufficiently high enough, or was the map quality high enough to resolve the lipid head groups? What were the distributions of these lipids? And do they play any role in the function of complex I?

We agree with the reviewer that the CxI-ND structures did not change significantly in comparison to the detergent-solubilised structures; we have elaborated on the minor differences observed in the Supplementary Discussion, and also added a section on page 8 to highlight increased order in subunit NDUFA9 in the deactive state that we tentatively attribute to the nanodisc (rather than detergent environment): *For the deactive-state structures, we were able to model substantially more of subunit NDUFA9 than has previously been possible in detergent-solubilised deactive/open structures (ref. 3,9,10,13,20,25). Only residues 324-331, adjacent to the disordered loop in ND3, were not modelled, suggesting that the nanodisc environment is able to contain the disorder that further propagates to residues ca. 186-195, 253-278 and 323-334 (ref. 13,20,25) in the detergent-solubilised enzyme. It is thus possible that structural changes to NDUFA9 in the membrane-bound deactive state are less extensive than previously supposed.*

With respect to the newly-resolved lipids, most of them reside on the periphery of the protein (i.e. at the protein-lipid interface along the membrane arm of complex I), next to regions of complex I that did not show any substantial structural changes and not between or within complex I subunits, so it was not possible to elucidate their roles beyond the stabilisation of protein regions in contact with the lipid bilayer. Following this question from the reviewer we have returned to our models and have extended our comment on the number and conservation of the phospholipids: we now note (page 8) that *As expected, none of the three DDMs modelled in the reference structure (ref. 40) were observed in CxI-NDs while the total number of phospholipids observed has increased to 47, including all 22 that were modelled in the reference structure (ref. 40).* The number of 'new' phospholipids has increased as we are now counting the total number observed across all models, which we consider, together with a more clear statement on the conservation, gives a more complete picture. The increased number is likely a combination of the greater resolution and map quality of our data ('up' to 2.3 Å resolution) with the lipid-filled nanodisc environment.

The synthetic phospholipids and their ratios for reconstitution were based on the known mammalian mitochondrial lipid composition, and imported here from established work on complex I-reconstituted proteoliposomes, notably Biner et al. 2020 (ref. 35) in which the lipid ratios were optimised and the complex I was shown to be highly active in proteoliposomes that were well coupled and capable of maintaining a substantial Δp to drive ATP synthesis by F₁F₀-ATP synthase. Finally, unless the lipid headgroups were surrounded by protein, it was usually not possible to resolve their specific identity due to their dynamic nature. As stated in the Supplementary Fig. 6 legend and now reiterated in the methods on page 39: *all non-cardiolipin phospholipids were modelled as phosphatidylethanolamines unless density features indicated phosphatidylcholine to be more likely*". We are therefore unable to comment further on the lipid distribution.

2. It could be helpful to show the chemical or protein components in the analyzed sample, which provides confidence in modeling the molecules in the cryo-EM densities. Mass spectrometry or LC-MS/MS may provide another layer of molecular evidence in modeling unknown densities in this study.

We thank the reviewer for this suggestion, but we are confident in the identities of all the protein components of our system, and have modelled all the densities attributable to protein moieties in our structures (other than from the nanodisc belt). No unknown protein densities were identified using the *Find blob* tool in Coot. Furthermore, the non-protein composition of our sample is known in detail. For these reasons we (unfortunately) regard it as unlikely that MS or LC-MS/MS will add more confidence to the modelling of specific unknown densities.

Minor concerns:

1. It is strongly recommended not to present a specified protein residue by superscripting the protein name, such as Tyr108^{NDUFS2}. 'NDUFS2-Tyr108', 'NDUFS2-Y108', or 'Tyr108 of the NDUFS2' is most seen. These changes can dramatically improve readability.

Thank you for this suggestion. However, we believe the current notation (superscripted subunit names) helps when there is a need to list various residues from different subunits (eg. His59^{NDUFS2}, Tyr108^{NDUFS2}, Thr156^{NDUFS2}, Met70^{NDUFS7} and Ser205^{ND1}) and when describing specific distances/interactions (eg. His59^{NDUFS2}-N_{ε2}-Asp160^{NDUFS2}-C_γ). We note that this notation has previously been used in complex I papers published in Nature Communications such as Yoga et al., 2020 (DOI: 10.1038/s41467-020-19778-7).

2. INTRODUCTION (paragraph 2): "... Cryo-EM studies of mammalian complex I have also identified a third state of particles that lack key regions of density, which ..."

 "... Cryo-EM studies of mammalian complex I have also identified a density of the third state that lacks the information of the key regions, which ..."

We apologise if this was not clear and have amended it to read: *Cryo-EM studies of mammalian complex I have also identified a third state, for which the density map lacks information in key regions of the enzyme, which...*

3. INTRODUCTION (paragraph 3): "... As expected, the structure shows a fully-bound Q₁₀ spanning the entirety of the channel, but, consistent with the observations of the short-chain substrate ..."

We prefer to write *...shows the fully-bound Q₁₀ spanning the entirety of the channel...* because the preceding sentence already mentions the same fully-bound Q₁₀ molecule.

4. INTRODUCTION (paragraph 4): "... By comparing the structures of substrate/ligand-bound and apo (substrate/ligand-free) forms ..."

We have removed the hyphen in front of "apo".

5. The author described that the intact nanodisc-reconstituted CxI-NDs displayed little piericidin-sensitive NADH:dQ activity. Would it be possible to analyze whether the reconstituted lipid or previously used detergent plays any role in function? Also, it seems that the active site is away from the lipid bilayer, and how would the bilayer reconstitution interfere with the dQ entering or function?

Our previous studies of isolated complex I reconstituted into proteoliposomes (ref. 33 and 35, also Jones et al 2016. DOI: 10.1002/anie.201507332) have demonstrated consistent NADH:dQ and NADH:Q₁₀ activities between the membrane-bound, detergent-solubilised and lipid-reconstituted complex, showing that the neither the preparation of the enzyme in detergent, the subsequent removal of the detergent for reconstitution, and/or the replenishment by lipids, affect the function of complex I. As noted, for example on page 12, DDM has been tentatively suggested to stabilise the deactive state (ref. 23,43) and our data suggest one possible explanation (although the functional consequences of adventitious DDM binding in the site remain unclear). Finally, although the 'reactive site' itself is ~20 Å above from the lipid bilayer, the entrance of the Q-binding channel (by which the reactive site must be accessed) is in the lipid bilayer – dQ, as well as Q₁₀, must enter from within the bilayer.

6. To completely remove the effect from the DDM detergent, has the author tested using BioBeads to ensure the residual detergent is minimal?

We thank the reviewer for this helpful suggestion. We plan to investigate how best to eliminate DDM for our future studies, although we did not test BioBeads here because the long incubation times required (at least several hours) were found to be detrimental to the kinetic activity of the enzyme in our earlier studies of reconstitution into proteoliposomes.

7. RESULTS (Resolution of three major classes of CxI-ND particles, paragraph 2): "... As reported previously, the state 3 map (see Supplementary Fig. 5) lacks clear densities for the C-terminal half of the ND5 transverse helix and ..."

We have amended "density" to "densities" accordingly.

8. It is recommended to use colors that are friendly to the color-blinded, especially the colors in Fig. 1. Thank you for this suggestion. We attempted to stick to colour-blind-friendly colours for all of our figures (such as the local resolution maps in Supplementary Fig. 4) but because complex I has so many distinct subunits, it was difficult to do so for Fig. 1. Fortunately, however, the subunit colours in Fig. 1 are not necessary for the understanding of the manuscript.

9. FIGURE 1: It would be appreciated if the author could show the views of the model or the density in orthogonal directions. Ideally, the membrane plane will be perpendicular or parallel to the viewing directions. Figure 1a shows the maps in a side-view (i.e. membrane plane parallel to viewing direction; left) and in a top-down-view (i.e. membrane plane perpendicular to viewing direction; right). These two figures are therefore in orthogonal directions. The view of the model (Fig. 1b) is tilted in a specific bottom/side-up view to best show the overlapping helices.

10. FIGURE 2 AND 5: Please label the distances for potential hydrogen bonds or electrostatic interactions. We have tested out attempts to show the distances but unfortunately these attempts were not successful because the figures are already crowded – the addition of the numbers detracts from the clarity and does not contribute to the understanding of the figure. We note that we have addressed the most critical distances (e.g. between Q₁₀ and Tyr108^{NDUFS2} or His59^{NDUFS2}) in the main text ... *The His sidechain forms a hydrogen bond (H-bond) with the 3-methoxy of the Q-headgroup (3.2 Å), rather than with either of the reactive carbonyls, which are in geometrically unfavourable positions (Fig. 2a). The Tyr sidechain is too distant (>4.3 Å) from the headgroup for a H-bond....* and also that any additional distances required will be readily accessible from the coordinate files.

11. FIGURE 2 AND 6: Please describe the green surfaces in the panels in figure legends. Thank you for bringing this omission to our attention. We have amended the respective figure legends to describe the green surfaces in Fig. 2 (...and ND1 cavity (the green surfaces identified by CASTp) are shown...) and Fig. 6 (Cavities (green surfaces) were identified by CASTp...).

12. FIGURE 4: It is recommended to show the green cavities in a closed form. The open cavities depicted in Fig. 4c and 4e are open to the matrix and therefore it is not possible to show them in a closed form.

13. Like a cholesterol molecule, the cholate molecule has α (smooth) face and β (spiky) faces, featuring its interactions with specific residues. For example, the α face prefers to interact with aromatic residues, whereas the β face favors the residues of Ile, Val, and Leu. When modeling the cholate coordinates into the cryo-EM density, do the authors find a similar pattern of interactions?

Thank you for this interesting point. The cholate molecule (in both orientations) is largely sandwiched between the aromatic rings of Phe224^{ND1} and Trp46^{NDUFS7}, which is likely to be why there is not a strong preference for one orientation over the other in the State 3 map. In orientation 2 (which has a marginally higher model-map CC_{mask} value), we do observe two Val adjacent to the β (spiky) face, but neither makes a substantial contribution to the interaction.

14. METHODS (Complex I reconstitution into nanodiscs; paragraph 1): "... and total concentration of 25 mg mL⁻¹ were each mixed with 200 nmol ..."

The closing bracket “)” after 25 mg mL⁻¹ was not removed as there is a preceding opening bracket “(” two lines before it.

15. The nominal magnification described in ‘Cryo-EM grid preparation and image acquisition’ of the Method section did not seem correctly calculated using a 5-micrometer pixel size of the Gatan K3 DED camera. Also, in the reported Table 1, the magnification was reported as 130,000X (affected by the post-column energy filtering), which did not match the description in the Method section. The author may need to correct one of them to make it consistent.

Thank you very much for bringing this mistake to our attention. We have amended the value in the table to 81,000x nominal magnification, which, under super-resolution electron counting mode, corresponds to 0.535 Å pixel⁻¹.

16. The author will need to clarify further how the pixel size in the density map was calibrated (0.7496 Å/pixel) from the measured (0.535 Å) (Table 1). When the calibrated pixel size was used, was the CTF estimation or refinement re-performed? Was the model density re-refined and re-adjusted to the correct scaling for individual Fourier pixels?

We apologise the lack of clarity. The data were processed with a nominal counting pixel size of 1.07 Å pixel⁻¹, polished at a nominal pixel size of 0.7523 Å pixel⁻¹, and then corrected to a calibrated pixel size of 0.7496 Å pixel⁻¹ at the postprocessing and local resolution stages. CTF estimation/refinement was not re-performed once the pixel size discrepancy was identified. The PDB models were refined against the re-scaled maps. We have amended the ‘Cryo-EM data processing’ section in the methods to include this information: *[particles] were then signal reverted to include the nanodisc densities, and repolished at 0.7523 Å pixel⁻¹ (1.4x bin)...* and later, *All data processing were done at the nominal pixel size of 0.7523 Å pixel⁻¹, and corrected to a calibrated pixel size of 0.7496 Å pixel⁻¹ at the post-processing or local resolution stages in RELION...*

17. METHOD(Cryo-EM data processing; paragraph 1): Please rewrite the second sentence, which was not clearly written. The name of the ‘MotionCorr’ is referred to another program for motion correction, which should not be written as “RELION’s implementation MotionCorr”. The amplitude contrast is used in contrast transfer function estimation, which should be separated into an independent sentence from what was described for the motion correction.

Thank you for these suggestions, we have amended the sentence as follows: *The micrographs were motion-corrected using RELION’s implementation of motion correction with 5 x 5 patches, and contrast transfer function (CTF) estimated using CTFFIND-4.1 (ref. 51) with an amplitude contrast of 0.1 and ResMax set to 5 Å.*

18. METHOD(Cryo-EM data processing; paragraph 1): What is the 3D reference used for automated particle picking? Was there any pre-treatment on the 3D reference?

The 3.1 Å DDM-solubilised complex I map (EMD-14127) was used as the 3D reference as noted by the reference in the text. The 3D reference was low-pass filtered to 20 Å. We have amended our sentence and added the EMD code for clarity: *...RELION’s AutoPicking tool with a 3D map input (EMD-14127) (ref. 40) low-pass filtered to 20 Å.*

19. METHOD(Cryo-EM data processing; paragraph 1): The sentence mentions the signal subtraction, which was performed to remove most of the non-complex I contribution. What is the non-complex I contribution and where are they?

We apologise for the lack of clarity. The non-complex I contributions refer to the densities attributed to the MSP2N2 nanodisc belt and the lipid bilayer within the nanodisc. We now explain: *Signal subtraction was then performed to remove most of the non-complex I contribution (MSP2N2 nanodisc belt and the lipid bilayer within the nanodisc) using the complex I mask.* The description of how the nanodisc mask was

made has now been repositioned at the end of the 'Cryo-EM data processing' section: *To make the nanodisc mask, a tight complex I mask was first generated from a working model using RELION MaskCreate, and subtracted from the consensus map to obtain densities for the nanodisc; leftover complex I densities were removed using the Map Eraser tool in UCSF ChimeraX (ref. 50), and the complex I-subtracted map then used as the RELION MaskCreate input.*

20. It is recommended not to present individual classes as C1, C2, C6, etc. A general description of 'class #' will be sufficient.

We have amended all mentions of 'C#' in the text and figures to 'class #'.

21. It may be a concern to use 'masking' and 'solvent flattening' interchangeably in the Method section, although the purposes of the two processing procedures are similar.

Thank you for raising this point. We have checked our use of these terms and adjusted the text on page 36 to now say: *At this early stage, 358,326 particles were 3D refined with a complex I mask (generated from a working model using RELION MaskCreate) and with solvent flattening to give a global resolution of 2.28 Å (according to a gold-standard Fourier shell correlation (FSC) of 0.143).*

22. It seemed that the ligand-bound state (2.30 Å resolution, 235,957 particles) has the largest particle population among all the others. Does it provide any implication in the energy landscape of the complex? Would it be possible to project the state onto the MD trajectory?

Although it is tempting to use the relative particle population as a proxy of the relative stability (or relative free-energy difference) between states we try to resist placing too much emphasis on these interpretations because sample preparation and bias in image picking or selection may artificially change relative particle populations. For example, states that are unstable during grid freezing may be decreased (or lost from the analysis entirely). Regarding projections of states onto the MD trajectory, we have done this in so far as it is possible in Fig. 3 and Supplementary Fig. 8 (shown by the star and bullet symbols). However, our MD simulations capture only 'active' configurations and thereby do not extend to the particular (deactive) ligand-bound state referred to.

23. It is recommended to show the quality of the model fitting, especially the densities relevant to the function, in Supplementary Data.

We have added a new Supplementary Figure to the manuscript to show the quality of the model fitting. Accordingly, the main text has been amended to reference the new Supplementary Figure, and subsequent supplementary figure numbering corrected. The figure legend is as follows: *Supplementary Fig. 5: Densities in the CxI-ND maps for key elements that differ between the states and representative densities for water molecules. Densities for the (a) ND1-TMH5-6 loop (residues 194-217), (b) NDUFS2-β1-β2 loop (residues 52-62), (c) NDUFS7 residues 47-51 and 74-83, (d) ND3-TMH1-2 loop (residues 24-55), (e) ND6-TMH3 (residues 49-76), and (f) representative water molecules in the deactive-ligand state (at the interface between NDUFS2 (beige), NDUFS3 (pale green), and NDUFS8 (green)). All map densities are shown at map thresholds of 4.5 in UCSF ChimeraX (ref. 7). Colours for the map densities are as follows: purple (active-Q10), salmon (active-apo), green (deactive-ligand), olive (deactive-apo), and turquoise (state 3).*

24. The authors modeled the atomic coordinates of the water molecules in multiple states. Are the coordinates of the waters consistent across these different structures? How are the water distributions in the bilayer with protein and outside the nanodisc? It would be helpful to look at the models along with their cryo-EM densities.

Whilst differences in resolutions make comparisons of water molecules tricky between the multiple states, many waters, where resolvable and in structurally conserved protein regions in the peripheral and membrane arms, are consistent across the active and deactive states (state 3 has no modelled waters). The numbers of water modelled in the active and deactive states are high (1,024 to 2,774, as given in Table

1) and, as expected, a majority of them are modelled outside the bilayer, especially in the peripheral arm. Limited water molecules are observed in the membrane domain, including along the hydrophilic axis and in the E-channel. Fig. 6 focuses on the differences to the E-channel between states, including differences in the water molecules. Finally, to illustrate the densities we have been sufficiently confident to model as waters we have added a set of representative water densities in a new Supplementary Fig. 5 (panel f).

25. It is recommended to add 3DFSC plots for individual structures to show whether the resolutions were determined and measured isotropically or anisotropically.

Thank you for this suggestion. We have added 3DFSC histogram plots for individual structures in Supplementary Fig. 4, and amended the 'Cryo-EM data processing' section in the methods and the figure legend: *the degree of directional resolution anisotropy [was] calculated using the 3DFSC program suite (ref. 52) (Supplementary Fig. 4).*

26. SUPPLEMENTARY FIG. 2: It is recommended to show the scale of the 2D class averages.

Thank you for this suggestion. We have added a scale bar for the 2D class averages.

27. SUPPLEMENTARY FIG. 4: It is recommended to show red (low) to blue (high) for local resolutions.

Thank you for this suggestion but we chose the colours in the local resolution maps in Supplementary Fig. 4 because they are colourblind-friendly.

Reviewer #4

The manuscript by Chung et al. reports a cryo-EM study of a mammalian complex I enzyme reconstituted into phospholipid nanodiscs with exogenous Q10 combined with enhanced sampling molecular dynamics simulations. Five distinct cryo-EM structures in different functional states are resolved at 2.3-3.0 Å resolution including one with a Q10 molecule occupying the full length of the Q-binding channel. These structures combined with molecular dynamics based free energy calculations provide a mechanistic picture on how substrate binding restructures the deactive Q-binding site.

In my opinion, this is an interesting study and has some important mechanistic implications in terms of Complex I ubiquinone interactions and the role of protonation states of D160 and H59 in these interactions. I only have a few comments that need to be addressed by the authors to clarify and improve the manuscript: We thank the reviewer for this positive evaluation.

1. As the authors also state both 'active' and 'deactive' resting states of Complex I have previously been characterized by cryo-EM. There is also a recent cry-EM structure of Q10-bound Complex I that the manuscript refers to in several places. It seems to me the distinction the authors try to make between their work and previously published work (in terms of their conclusions and not methodology) relies more on the MD side of the study. My question is why they only use one of their 5 cryo-EM structures to do MD and more specifically why they use the apo active state rather than the active-Q10 state and insert the Q10 molecule instead. Since MD is really what distinguishes this work from the previous ones, perhaps, they should expand more on this side of their study.

First, we appreciate this comment, but contend that the new information in our manuscript is by no means restricted to the MD simulations. In addition to the deactive (and state 3) cryo-EM structures presented, which have not been described previously with specific ligand occupation or related conformational changes in the Q-binding site loops, the position of the substrate headgroup (and nearby residues) presented in our active-Q₁₀ structure differs significantly to those described in previous studies, including the recent porcine structure of Q₁₀-bound complex I (Gu *et al.* 2022). Our MD work builds upon our observed mode of substrate binding, adding new insights into, for example, the protonation states of the environment that we propose stabilise the binding pose. Second, we 'inserted' the Q₁₀ into an apo structure

for modelling so that our simulations would not be biased by starting from the observed binding pose. Our Q₁₀-inserted MD model is a thereby true 'prediction' - the excellent agreement we observed between it and our Q₁₀-bound cryo-EM model is striking, and this reveals the start of a fundamental understanding of how Q₁₀ binding is stabilised in our unusual binding channel. Finally, simulations of the deactive state were not the focus of this work – which aimed to use MD simulations to predict and define the binding poses of Q₁₀ in the Q-binding site of active-state complex I. Using MD simulations to approach the deactive state and the active/deactive transition of complex I is a separate challenge, involving consideration of partially incomplete structures (intrinsically disordered loops in NDUFS2, ND3 and ND1, large numbers of titratable residues with uncertain protonation states) and is beyond the scope and remit of this current paper.

2. A methodological aspect of this work that distinguishes it from Ref. 27, for instance, is the use of nanodiscs. The authors state in the Introduction that "the nanodiscs provide a native membrane-like environment and eliminate potential artifacts from the detergent micelle typically present in cryo-EM analyses". However, in the Results section, there is no evidence of any differences. I think this needs to be clarified. There is a bit of a disconnect between the Introduction and Results.

We thank the reviewer for bringing this to our attention. We originally developed the nanodisc approach to enhance the chance of observing Q₁₀ in the channel of the mammalian enzyme – a feat that has not been possible in the detergent-solubilised isolated enzyme, and that was a serendipitous observation in the supercomplex-bound enzyme – as well as to eliminate potential detergent artefacts. Indeed, it was important to confirm that detergent-artefacts are not substantial. However, we accept the reviewers point and we have now expanded our discussion of the differences between CxI-ND and DDM-solubilised complex I on page 8. In this section we also draw proper attention to the increased order in subunit NDUFA9 in the deactive state in the nanodisc-bound enzyme (which we have reconsidered as a result of the review process): *Comparison of the CxI-ND active-state structure with a DDM-solubilised active-state bovine structure [Protein Data Bank (PDB) ID: 7QSD; Electron Microscopy Data Bank (EMDB) ID: 14127] (ref. 40) revealed no material differences, with convincing map-to-map correlation (0.97) and RMSD values (0.26-0.31 Å for the membrane-bound core subunits, 0.32-0.35 Å overall). For the deactive-state structures, we were able to model substantially more of subunit NDUFA9 than has previously been possible in detergent-solubilised deactive/open structures (ref. 3,9,10,13,20,25). Only residues 324-331, adjacent to the disordered loop in ND3, were not modelled, suggesting that the nanodisc environment is able to contain the disorder that further propagates to residues ca. 186-195, 253-278 and 323-334 (13,20,25) in the detergent-solubilised enzyme. It is thus possible that structural changes to NDUFA9 in the membrane-bound deactive state are less extensive than previously supposed.*

3. I am somewhat surprised by the long restraining stage in the MD simulations, which is longer than the production run if I read the Methods section correctly. Is there a reason the authors needed such long restraining stage?

There are a few reasons for running a long restraining stage. First, we had to be confident that lipid distribution in the mixed membrane and bound to protein, as well as internal solvation – particularly in the Q-channel – were both equilibrated. This may require a long simulation time (~400 ns), as noted in the manuscript. Second, we started from an apo structure, so an additional equilibration stage was necessary after inclusion of Q₁₀ in the Q-channel and reassignment of the charges in reactive groups (His59^{NDUFS2} and Asp160^{NDUFS2}). In both stages, restraints in heavy protein atoms were necessary to avoid artefactual structural transitions. Production simulations for each charge state were also rather long: 300 ns for canonical MD and 140 ns for metadynamics MD (an enhanced sampling method) and we consider these appropriate for conformational sampling in confined regions such as the Q-channel.

4. I think the free energy calculation part needs proper error analysis. The authors state: "Convergence within ±1 kJ/mol of free energy differences in the CV profile was reached after 70 ns". However, the

apparent convergence does not necessarily give a good sense of error. A common technique used in the free energy calculation field is bootstrapping that can provide better estimates of the uncertainty.

We agree that error analysis is relevant and have now included an estimation of statistical uncertainty in the free-energy profiles of Fig. 3a, obtained with the bootstrap technique. The legend for this figure and the respective Methods section have been amended accordingly.

Reviewers' Comments:

Reviewer #2:

Remarks to the Author:

In their manuscript "Cryo-EM structures define ubiquinone-10 binding to mitochondrial complex I and conformational transitions accompanying Q-site occupancy" Chung et al. describe the structures of bovine (*Bos taurus*) mitochondrial complex I reconstituted into nanodiscs in the presence of the substrate ubiquinone-10 (Q10). The authors identify 5 distinct states of the complex present in their preparation and two distinct modes of Q10 binding to the active complex. They also identify an apo state of the active complex that lacks Q10, which is important as this state was proposed to not be present from a recent structural study of porcine respiratory supercomplexes (PMID: 35145322). From molecular dynamics simulations they conclude that the mode of Q10 binding observed in their structures represents a 'pre-reactive' state. Overall, the paper presents important new results on Q10 binding to mammalian mitochondrial CI, clarifies discussions around the observed distinct states of the complex and presents a new biochemical system in which to further study the mechanism of the energy-conversion by the complex. The manuscript is clearly written and will be of great interest to people in the field.

In their revised submission Chung et al have address all of my comments satisfactorily and I recommend publication of the manuscript.

Reviewer #3:

Remarks to the Author:

The authors have addressed all of my questions and improved the manuscript. I don't have any further suggestions. The manuscript looks very good to me now.

Reviewer #4:

Remarks to the Author:

The authors have addressed all my concerns.

Reviewer #2 (Remarks to the Author):

[...] In their revised submission Chung et al have address all of my comments satisfactorily and I recommend publication of the manuscript.

Reviewer #3 (Remarks to the Author):

The authors have addressed all of my questions and improved the manuscript. I don't have any further suggestions. The manuscript looks very good to me now.

Reviewer #4 (Remarks to the Author):

The authors have addressed all my concerns.

We thank all the reviewers for their constructive comments and positive evaluations of our work.